# Computing water flow through complex landscapes – Part 3: Fill-Spill-Merge: Flow routing in depression hierarchies

Richard Barnes[1,2,3], Kerry L. Callaghan[4,5], and Andrew D. Wickert[4,5]

[1]Energy & Resources Group (ERG), University of California, Berkeley, USA
[2]Electrical Engineering & Computer Science, University of California, Berkeley, USA
[3]Berkeley Institute for Data Science (BIDS), University of California, Berkeley, USA
[4]Department of Earth & Environmental Sciences, University of Minnesota, Minneapolis, USA
[5]Saint Anthony Falls Laboratory, University of Minnesota, Minneapolis, USA

**Correspondence:** Richard Barnes (richard.barnes@berkeley.edu)

**Abstract.** Depressions—inwardly-draining regions—are common to many landscapes. When there is sufficient moisture, depressions take the form of lakes and wetlands; otherwise, they may be dry. Hydrological flow models used in geomorphology, hydrology, planetary science, soil and water conservation, and other fields often eliminate depressions through filling or breaching; however, this can produce unrealistic results. Models that retain depressions, on the other hand, are often undesirably expensive to run. In previous work we began to address this by developing a depression hierarchy data structure to capture the full topographic complexity of depressions in a region. Here, we extend this work by presenting a Fill-Spill-Merge algorithm that utilizes our depression hierarchy data structure to rapidly process and distribute runoff. Runoff fills depressions, which then overflow and spill into their neighbors. If both a depression and its neighbor fill, they merge. We provide a detailed explanation of the algorithm as well as results from two sample study areas. In these case studies, the algorithm runs 90–2,600× faster (with a 2,000–63,000× reduction in compute time) than the commonly-used Jacobi iteration and produces a more accurate output. Complete, well-commented, open-source code with 97% test coverage is available on Github and Zenodo.

## 1 Introduction

Depressions (see Lindsay (2015) for a typology) are inwardly-draining regions of a DEM that lack any outlet to an ocean or other designated base elevation. Depressions occur naturally, and can be formed by glacial erosion and/or deposition (Breckenridge and Johnson, 2009), compressional and/or extensional tectonics (Reheis, 1999; Hilley and Strecker, 2005), and cratering (Cabrol and Grin, 1999). They often host lakes and wetlands by retaining water locally. Depressions may themselves contain depressions. Such regions confound algorithms for geomorphological and terrain analysis, as well as those for hydrological modeling, because many such algorithms simply route water down topographic slope following the local gradient: depressions neither fill with water, nor drain.

Many hydrological models deal with the complexity of depressions by removing them. This can be done either by filling the depressions with earth so that they form a flat region of landscape (e.g. Jenson and Domingue (1988); Martz and Jong (1988)); breaching (Martz and Garbrecht, 1998) or carving them (Soille et al., 2003) so that water flows from their lowest point through

the carved channel and onward to downstream regions; or some combination of these (Lindsay and Creed, 2005b; Schwanghart and Scherler, 2017; Soille, 2004; Lindsay, 2016). This approach is justified for situations in which spatiotemporal aspects of the analysis allow depressions to be ignored or for cases in which all depressions can be considered to be data errors (Lindsay and Creed, 2005a). Historically, many DEMs were constructed from sparse data, and small data errors produced depressions, especially in flat areas (O'Callaghan and Mark, 1984). Such an assumption is no longer justified, as improved and increasingly high-resolution data have become available (Li et al., 2011). Even coarse-resolution data are capable of resolving real-world depressions (e.g. Riddick et al., 2018; Wickert, 2016). With this in mind, new approaches are beginning to be examined, particularly in post-glacial landscapes where depressions have a significant impact on local hydrology (e.g., Lai and Anders, 2018) and therefore cannot be ignored during modeling.

FlowFill (Callaghan and Wickert, 2019) began to combat this problem by routing water across landscapes in a way that conserved water volume, creating flow-routing surfaces that could still contain real depressions. Under reasonable runoff conditions, their results show landscapes that still contain depressions and disrupted flow routes. The FlowFill method iteratively routes water from higher to lower terrain. As depressions fill, they pose an extreme challenge to such a method: since water seeks a level surface, the surface of a filled depression must eventually become flat and any fluid flowing onto the surface diffuses across it. Even for moderately-sized surfaces it can take many iterations for a solver to reach steady state; we provide a theoretical analysis of this in Section 4.1. Runtimes for FlowFill ranged from seconds to days: large datasets quickly became unwieldy. Of those examples tested by Callaghan and Wickert (2019), the slowest was a dataset of 4,176,000 cells which took approximately 33 hours for FlowFill to process. In contrast, the Fill-Spill-Merge algorithm presented here fills a similarly-sized dataset in 8.7 s.

Other authors have considered the problems of extracting nested depression hierarchies and dynamically routing water through them. However, these previous approaches are either slow, inexact, or both; additionally, most previous efforts were not accompanied by source code, limiting their utility. Barnes et al. (2020) provide a more thorough literature review which we briefly recap here. A hierarhical segmentation by Beucher (1994) did not produce a data structure on which flow could be routed. Salembier and Pardas (1994) generated a hierarchical segmentation by repeatedly simplifying source images; hydrologically speaking, this can lead to unacceptable degradation of terrain information. Arnold (2010) developed an algorithm similar to the one here, but without source code; the algorithm also generates looping topologies that require correction. Wu et al. (2015) and Wu and Lane (2016) constructed depression hierarchies by first smoothing a DEM and then extracting vector contour lines from it. Wu et al. (2018) build on this approach by discretizing the DEM into a number of horizontal slices. Both approaches sacrifice exactness and the latter requires $O(N^2)$ time. Cordonnier et al. (2018) use planar graph minimum spanning trees to construct a hierarchy of depressions, but do not produce a data structure water can be routed on. In contrast, the Fill-Spill-Merge algorithm relies on a well-defined data structure (Barnes et al., 2020); has complete, well-commented source code with extensive correctness tests (Barnes and Callaghan, 2019, 2020); has strong efficiency guarantees (§4.1) which are realized on actual and simulated datasets (§4.2); and provides exact answers.

To achieve this, we developed a data structure—the *depression hierarchy*—which represents the topologic and geographic structure of depressions. In an accompanying paper, we provide details concerning how a depression hierarchy is constructed (Barnes

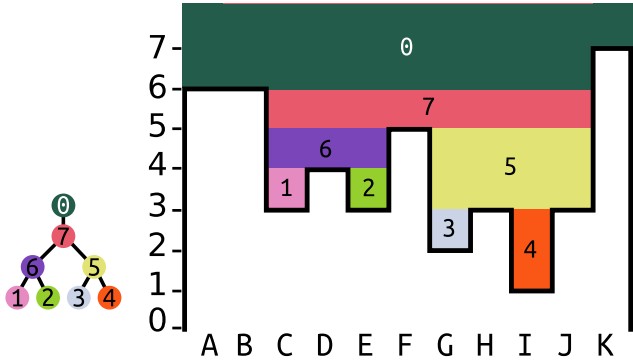

**Figure 1. A single subtree of a depression hierarchy and the depression it represents.** Depressions 1–4 are leaf depressions. Depression 6 is a parent depression (also termed a meta-depression) that contains depressions 1 and 2. Water from the plateau on the left above cells $A$ and $B$ might *fill* Depression 1 (cell $C$), causing it to *spill* into Depression 2 (cell $E$). Only when both depressions are full do they *merge* and begin filling Depression 6 (cells $C$, $D$, and $E$). Modified from Barnes et al. (2020).

et al., 2020). In this paper, we explain how a depression hierarchy can be leveraged to accelerate hydrological models using a paradigm we call *Fill-Spill-Merge*.

## 2 Using The Depression Hierarchy

Many of the techniques in this paper are based on binary tree data structures and their traversals. Although we define terms below, more complete explanations and visual examples can be found in the text for any introductory undergraduate course on data structures. We recommend Skiena (2008) and Sedgewick and Wayne (2011) as good references. In particular, a good understanding of recursion will be helpful.

### 2.1 Terminology

Depressions can themselves contain depressions, as shown in Figure 1. A depression hierarchy (DH) is a data structure representing a forest of binary trees, as shown in Figure 2a, that represents the relationships between depressions (Figure 2a–d). Each node in the DH represents a depression. Nodes higher in the DH are depressions that themselves contain depressions; we term these *meta-depressions*. Although the depression hierarchy could be generalized to n-ary trees using multiple flow direction routing, the binary simplification is sufficient to cover most use cases. A node in the DH can have several classifications:

- **Parent**: A node, such as #10 and #12 in Figure 2a, that represents a meta-depression, and whose topological descendants therefore also form depressions.

- **Child**: A depression, such as both #10 and #1 in Figure 2a, that geographically and topologically exists within the meta-depression formed by its parent.

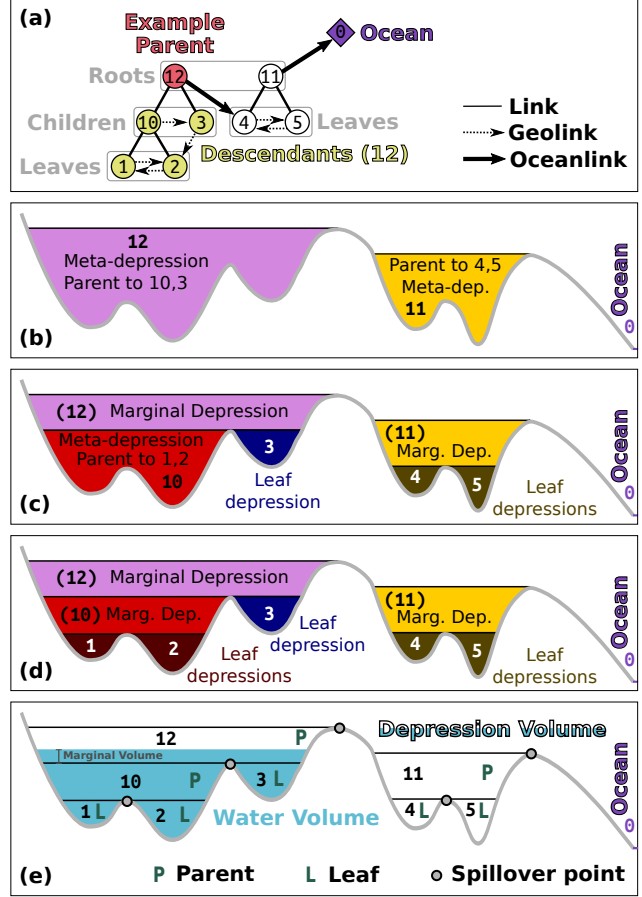

**Figure 2. Terminology for the depression hierarchy and water flow through it.** The depression hierarchy shown here is drawn from the left hand side of Figure 1 from the companion paper by Barnes et al. (2020). **(a)** Topology. A *parent* and its *descendants* are associated with depressions (b–d). Direct descendants are called *children*. *Leaves* are the terminal members of the depression hierarchy; they have no children and represent simple depressions (i.e., those that are not meta-depressions). Members of a single *binary tree* are joined in their hierarchy through *links*; directional links that represent water-spillover directions between geospatially adjacent depressions are called *geolinks*. Flow from one binary tree into another and towards the ocean follows the *oceanlinks*. Though only one binary tree is shown, the ocean may be the parent to an arbitrarily large *forest* of binary trees. **(b)** Parents in the hierarchy form *meta-depressions* — depressions that encompass other depressions. **(c)** These meta-depressions contain *leaf depressions* — depressions that themselves contain no depressions. These are associated with leaves in the depression hierarchy. Meta-depression 12 also contains another meta-depression, 10. The regions of Depressions 11 and 12 that lie above their child depressions are termed "marginal depressions". **(d)** Meta-depression 10 contains leaf depressions 1 and 2. **(e)** Using the depression hierarchy to simulate water flow. Water first fills *leaf depressions* before flooding into neighboring *depressions*. Once a depression and its neighbor are completely filled, their *parent* begins to flood. The *depression volume* is the full geometric volume of the depression. The *water volume*, naturally, is the volume of water within a given depression. The *marginal volume* is the volume of water partially filling the top-level meta-depression; appropriately spreading this water across the landscape is the topic of Section 3.3.

- **Leaf**: A depression, such as #1 and #2 in Figure 2a and Figure 2d, that has no children. The leaves of the binary trees represent the smallest, most deeply-nested depressions. If a landscape were initially devoid of water, then water flowing down slopes would begin to collect in some subset of these leaf depressions before it would begin to fill their parent depressions.

- **Root**: A depression, such as #0, #11, and #12 in Figure 2, that has no parent. This term may also refer to any node that is used as the starting point for a traversal that only considers the node and its descendants.

- **Descendant**: A child of a given parent, or the child of a child of that parent, and so on. In Figure 2a, #1, #2, #3, and #10 are all descendants of #12.

- **Sibling**: Every node has either no children (leaf nodes) or two children. Nodes which share a parent are siblings. In Figure 2a, #1 and #2 are siblings, as are #4 and #5.

As depressions fill, their water surfaces eventually reach a *spill elevation* (Figure 2e) at which they overflow into neighboring depressions. During this spilling, water flows from a depression into a geographically neighboring leaf depression, topologically connected by a *geolink*. The spill elevations in Figure 1 are the highest points of each band of color.

Each node in the DH is associated with several properties:

- **Depression volume**: This is the *total* volume of water that the depression, including all of its descendants, can contain before spilling over.

- **Water volume**: This is the *total* volume of water *actually being stored* in the depression. A parent depression will have a non-zero water volume only if both of its children are completely full and the parent itself contains some additional volume of water. In this case, the water volume will be the sum of the water volumes of the children and the additional margin of water contained within the parent (i.e., the "marginal volume" indicated on Figure 2e). Parents whose children are not both filled with water will have a water volume equal to zero. In this way, we can use this property to determine which portions of the DH are fully or partially filled, and which are the highest water-containing nodes in any of the binary trees.

- **Geolink**: When a depression spills, its water passes into the subtree rooted by its sibling. However, in a full model of flow, the water would move downslope from the spill cell into whichever leaf depression of the sibling is geographically proximal to the spill cell. *Geolinks* are pointers from depressions higher in the DH to the leaf depressions that receive their water if they overflow. These are the dashed lines shown in Figure 2a. Geolinks are similar to the connections used in a threaded binary tree (Fenner and Loizou, 1984).

- **Oceanlink**: Depressions high in the mountains may overflow down escarpments to depressions far below. In this case, the depressions do not overflow into each other: the relationship is one-way. There can be multiple such escarpments, so this can happen multiple times. In such cases, each group of depressions forms a proper binary tree. However, the root

of one of the trees has an *oceanlink* to a leaf node of the downstream binary tree. In Figure 2, both #11 and #12 are the

root nodes of a set of nested depressions. #12 has an oceanlink (heavy arrow) to #4, one of the leaf depressions of #11.

#11 itself has an oceanlink to the ocean. In many of the algorithms discussed below, oceanlinked nodes are processed

similarly to children.

Within the algorithm, oceanlinks and geolinks are used for different purposes: an oceanlink tells us that the depression in

question has grafted onto the leaf node of another tree of the depression hierarchy, locating a route for overflowing water to

eventually reach the ocean. The depression to which it is oceanlinked is considered its parent, but it is not the child of that

depression because water flows only one way along an oceanlink. In Figure 2a, depression #4 can be considered the parent

of #12, but #12 is not the child of #4. This is because #12 is not physically contained within #4, but #12 will send all of its

overflowing water to #4, as shown in Figure 2b–e. #4 will not contain the total water volume contained within #12, unlike other

parents. Geolinks route water within geographically adjacent depressions contained in the same meta-depression.

**2.2  Traversals**

With these linkages in place, we can consider various ways of traversing the trees. Given a binary tree $T$ with left and right

children $T.L$ and $T.R$, a breadth-first traversal considers both $T.L$ and $T.R$ before considering any of $T.L.L, T.L.R, T.R.L,$

or $T.R.R$. A depth-first traversal, on the other hand, will consider $T.L$ and all of its descendants before considering $T.R$ or any

of its descendants. The tree traversals we perform in this paper are all depth-first.

Depth-first traversals are most naturally expressed via recursion and come in three types: in-order, pre-order, and post-order.

Let a recursive traversal function be called $r(\cdot)$ and the processing we perform on a particular node in the tree $p(\cdot)$, then the

traversals are given by:

– in-order: $r(T.L)$ then $p(T)$ then $r(T.R)$

– pre-order: $p(T)$ then $r(T.L)$ then $r(T.R)$

– post-order: $r(T.L)$ then $r(T.R)$ then $p(T)$

**3  The Algorithm**

The Fill-Spill-Merge algorithm consists of several steps, outlined here, depicted in Figures 3 and 4, and shown in flowchart

form in Figure 5. This paper is also accompanied by complete, well-commented source code; the reader may find it helpful to

download this code and refer to it as an additional reference. First (§3.1), surface water needs to move downhill, either to the

ocean (i.e., a designated sink region or the map edge) or to collect in pit cells – the deepest points within leaf depressions. Note

that the landscape may already have standing water at this stage. This operation takes place across all the cells of the DEM.

Second (§3.2), water is redistributed across the depression hierarchy such that any depressions that have filled sufficiently spill

over into neighboring depressions and, if both depressions are full, flood their parent to merge into a single, larger body of

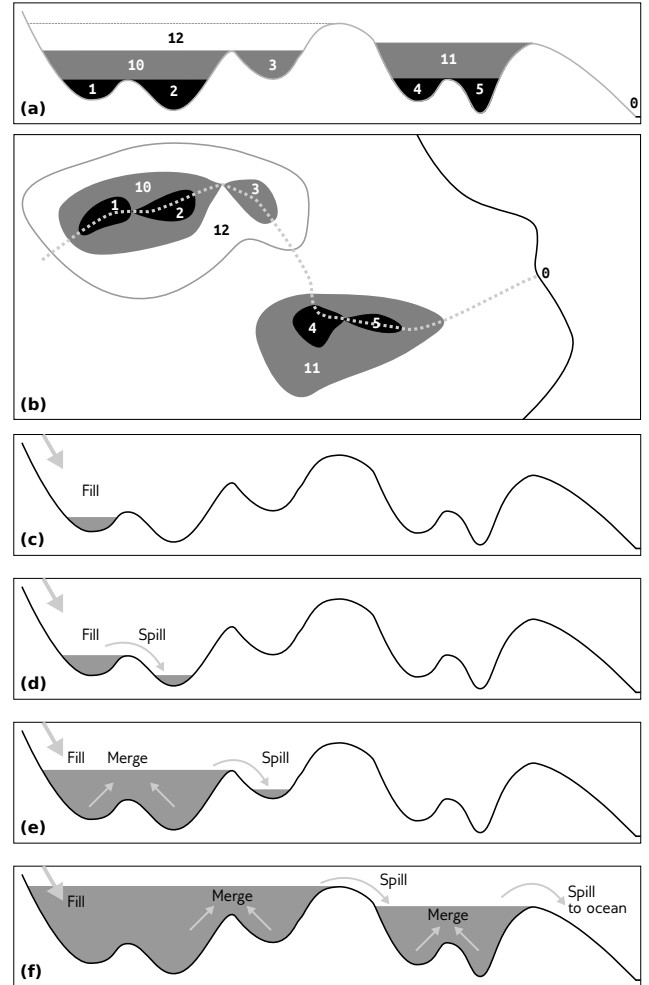

149

**Figure 3. Fill-Spill-Merge process.** Water moves through topographic depressions by filling them, spilling over sills, and merging to form meta-depressions. **(a)** Topographic cross section with labeled leaf depressions and their parents, following the left-hand side of the depression hierarchy in Figure 2. "0" represents the ocean; other numbers represent leaves and parents that together form depressions and meta-depressions. **(b)** Map showing this depression structure; the cross-section in (a) follows the dotted gray line. **(c)** A water source to the left begins to fill Depression 1. **(d)** Continued water input causes Depression 1 to overflow and spill into Depression 2. **(e)** Depression 2 fills, causing Depressions 1 and 2 to fill their parent (10) and merge to form a metadepression. This metadepression overflows into Depression 3. **(f)** Depression 3 fills and merges with Meta-Depression 10 (1 and 2 being implied members based on their position in the hierarchy) to flood their parent, 12. After Meta-Depression 12 overspills, it enters Depression 4, which then fills and spills into Depression 5. After Depression 5 floods, its waters join with those from Depression 4 to fill Meta-Depression 11, which then spills to the ocean. Figures 4 and 5 describe the algorithm in more specific detail.

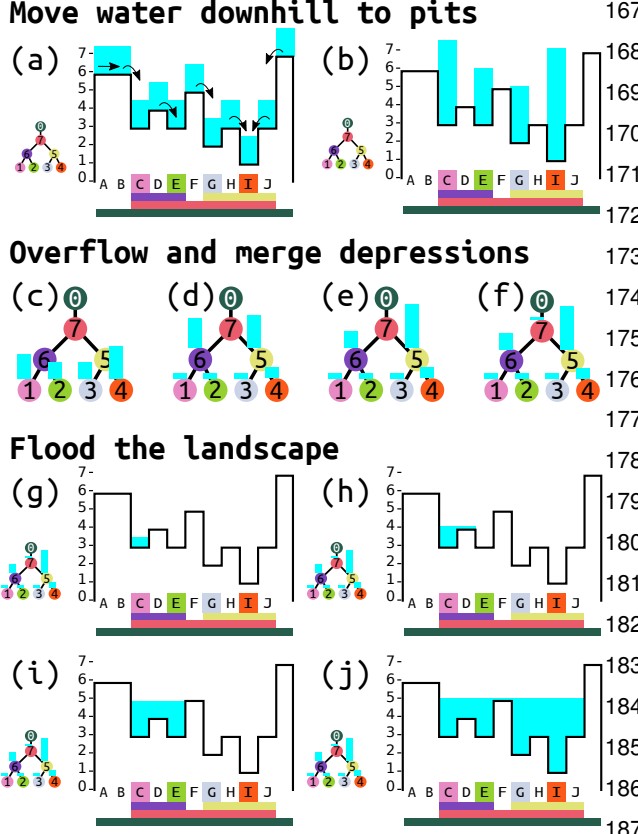

## Move water downhill to pits

(a) (b)

## Overflow and merge depressions

(c) (d) (e) (f)

## Flood the landscape

(g) (h)

(i) (j)

**Figure 4. Visual Overview of the Algorithm.** Black outlines represent the elevations of the cells. Blue areas are the heights of water in each cell or depression within the depression hierarchy. Capital letters label cells, and numbers on colored dots label depressions. Colors at the base of each panel match the colored dots and indicate to which depression each cell belongs. The algorithm consists of three major stages (Figure 5). From its initial distribution **(a)**, water is moved downhill following flow directions in the steepest downslope direction from each cell, as indicated by the arrows. Water continues to move downslope until it reaches the pit cells (**b**, §3.1). Water is then moved within the depression hierarchy (**c–f**, §3.2). **(c)** shows the initial distribution of water within the depression hierarchy, based on how much water was in the pit cell of each depression. Water in depressions with insufficient volume overflow first into their sibling depressions and then – if the sibling depression becomes filled – passes to their parents. All of the leaf depressions in (c) are completely filled, so no sibling depressions can accommodate more water. Therefore, depressions 1 and 2 pass their overflowing water up to their parent, depression 6, and depressions 3 and 4 pass their overflowing water up to their parent, depression 5. **(d)** Depression 6 is now overflowing, but its sibling, depression 5, is not full, so depression 6 passes as much of its overflowing water as it can to depression 5. **(e)** Once depression 5 is full, some overflowing water still remains, so this is passed to the parent, depression 7. **(f)** In this case, depression 7 is able to accommodate the remainder of the water. Had depression 7 also overflowed, the leftover water would have overflowed into the ocean and been disregarded. Depressions to be flooded are then identified and flooded (§3.3). Since depression 7 contains water, we know that all of its descendants must be completely full. Therefore, we can flood these all at the same time, on the level of depression 7. Any one of the pit cells within depression 7 is arbitrarily selected as the starting point **(g)**. More cells are added until all of the water has been accommodated. **(h–j)** are a visual representation of this process, although the algorithm would first locate affected cells C–J, and then calculate the final height of water in all of these cells in a single step.

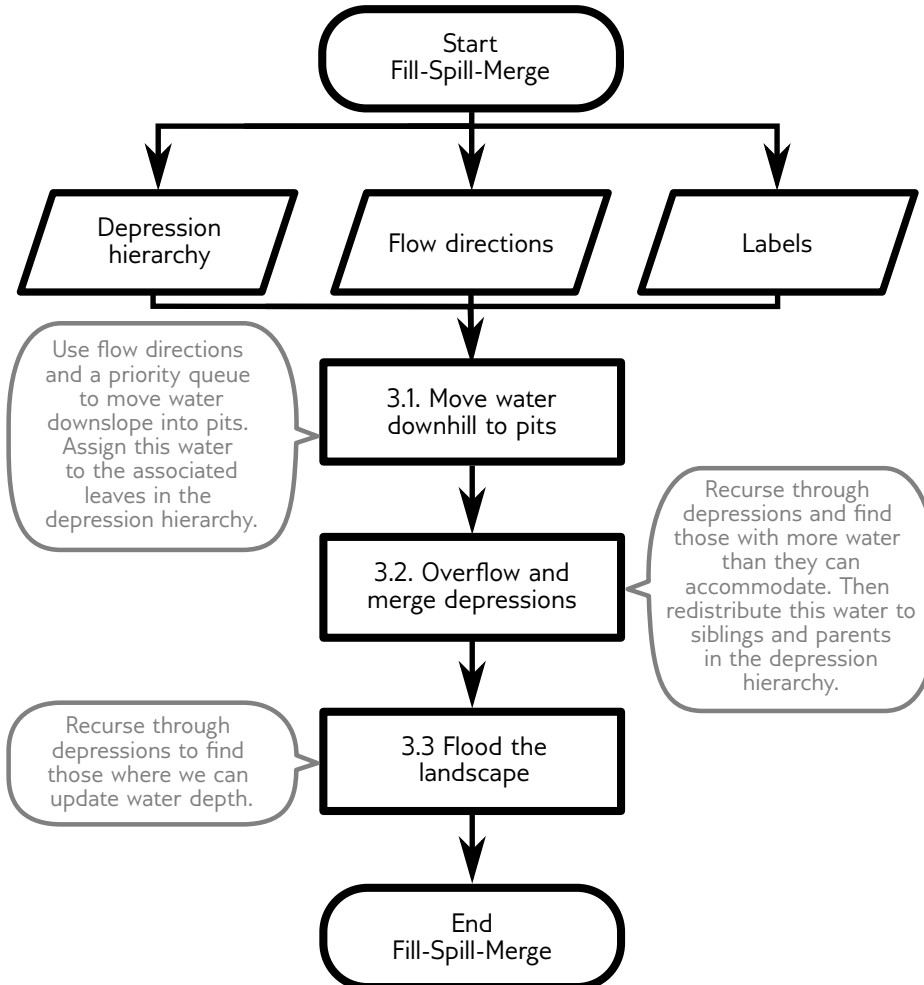

**Figure 5. Flowchart showing the main steps taken by the algorithm.** These steps are described in more detail in §3.1 to §3.3.

water within a meta-depression. This operation is done without explicitly considering the cells of the DEM, which makes it very fast. Third and finally (§3.3), the water within the depression hierarchy is translated into an extent and depth of flooding across the topographic surface (DEM).

Computing a depression hierarchy (Barnes et al., 2020) is a necessary precursor to running Fill-Spill-Merge. The specific outputs from the depression hierarchy algorithm that are used in the Fill-Spill-Merge algorithm are:

– *DH*: the depression hierarchy itself.

– *Flowdirs*: a matrix of flow directions, indicating which of a cell's neighbors receives its flow. Because Priority-Flood (Barnes et al., 2014) is used to generate the depression hierarchy, flat areas are automatically resolved.

– *Labels*: a matrix indicating the leaf depression to which each cell belongs.

By routing water according to the DH, we significantly accelerate the compute speed and ensure that the full network of
depressions is a topologically correct directed tree. Each of the following subsections details one of the numbered steps along
the central path of the flowchart shown in Figure 5.

## 3.1    Move Water Downhill to Pits

We route water in a similar way to standard flow-accumulation algorithms (Mark, 1988; Wallis et al., 2009; Barnes, 2017), but
for completeness summarize our approach here. Flow directions for each cell have already been identified by the depression
hierarchy algorithm. Each cell calculates how many of its neighbors flow into it. We call this value the cell's dependency count,
as it describes the number of immediate upstream cells whose flow accumulation must be resolved before flow accumulation
at the given cell can be computed. Local maxima in the DEM are identified as those cells that receive no flow from any
neighbor. These local maxima are placed in a queue. Cells are then popped (i.e., noted while being removed) from this queue.
The cells determine how much flow they generate locally (perhaps referring to a matrix of rainfall values, but also including
existing stores of standing water) and add this to their flow accumulation value. They then add their flow accumulation to
their downstream neighbor's and set their own flow accumulation value to zero. The neighbor's dependency count is then
decremented. If the neighbor's dependency count has reached zero during this step, it is added to the end of the queue. This
process of accumulating flow, passing it downstream, decrementing the dependency count, and adding cells to the queue
continues until the queue is empty, at which point every cell on the map has been visited and any water has been moved
downslope. Braun and Willett (2013) present an alternative formulation based on a depth-first traversal, but Barnes (2019)
demonstrates that a breadth-first ordering, such as that presented here, is better suited to parallelism.

When the accumulated flow reaches the pit cell of a depression, the downhill-directed flow routing stops because there is no
downhill neighbor to receive the flow. At this point, all of the flow-accumulated water in the pit cell is moved into the pit cell's
associated leaf depression in the DH. That is, the water is moved out of the geographic space and into the topologic space. This
then enables mass-conserving depression flooding via rapid Fill-Spill-Merge calculations, as detailed below.

## 3.2    Overflow and Merge Depressions

At this point, the Fill-Spill-Merge algorithm has routed all of the surface water into either the ocean or into the leaf nodes of the
DH. The next step is to redistribute this water through the DH to nodes with enough volume to contain the water, and to send
any excess water to the ocean. This set of operations can be performed entirely in the depression hierarchy without reference
to the digital elevation model.

Intuitively, the process of filling, spilling, and merging can be visualized as occurring from leaf nodes to their parents
(Figure 3). When a leaf depression initially contains more water than it can hold, the water will be redistributed by spilling
over into the neighboring depression. If this neighboring depression is already full, then the excess water must pass to the
parent of both the depression and its neighbor. This process continues recursively until either the supplied water is exhausted
or this water reaches the ultimate parent, the ocean. In this latter case, all excess water is dropped from the model and the ocean
is unaffected.

To effect the intuition developed above, we need a well-defined way to visit all of the nodes in the depression hierarchy. A post-order traversal allows us to visit both of a node's children and their descendants before calculating any quantities on the node itself. The result is that leaves get processed before their parents. However, a single traversal is insufficient: we need one traversal (the "outer" traversal) to identify nodes that have excess water and another traversal (the "inner traversal") to distribute this water. The outer traversal may launch the inner traversal many times as it works its way up hierarchy. Pseudocode showing these travels is available in §6.1 and §6.2.

To efficiently redistribute water, the Fill-Spill-Merge algorithm performs nested depth-first traversals of the DH. The outer traversal (§6.1) is post-order and considers each meta-depression in turn, from the most deeply nested to the least. For each meta-depression, an inner traversal (§6.2) handles its overflows by moving water to its sibling (starting by filling the sibling's descendants) and, if there's any left, passing it to the depression's parent. In this way, the outer traversal maintains an invariant (a property which is true before and after each call a function): any meta-depression it has processed does not contain an overflow. Put another way, the outer traversal finds problems and the inner traversal fixes them.

The outer traversal of the DH (which is, after all, a forest of binary trees) begins with the ocean. For each depression, the algorithm first recurses into its oceanlinks, if any, and then into the left and then right child. In the post-order portion of the traversal (which starts from the leaves and moves back up through the depression hierarchy), the algorithm identifies any depressions containing more water than they can accommodate. This process continues until the recursion returns to the ocean, at which point any additional water is assumed to be added to the ocean without impacting sea level, though this total discharge to the sea is recorded within the "ocean" depression.

When an overfilled depression is located by the outer traversal above, its water needs to be redistributed to neighbouring depressions. If we call the overfilled depression $D$, then the water can be redistributed by starting a second, inner post-order traversal at $D$. This inner traversal carries Excess Water from one depression to another until it has found a home for all of it. When we pass water into a depression, it can go to one of three places: the depression itself, its sibling, or its parent. Distributing the water to any of these places may itself cause an overflow. Therefore, the inner (pre-order) traversal comprises the following steps:

1. Call the depression that we are currently considering $B$. This may be the depression we originally considered, depression $D$, or it may be some other depression reached during the steps detailed below. If $B$ is overflowing, we add the overflow to the Excess Water the inner traversal is carrying. If $B$ has spare capacity we add water from the Excess to $B$ until either it fills or all of the Excess Water the inner traversal is carrying is used.

2. At this point, the inner traversal can terminate if: (i) there is no water left, (ii) $B$ is the parent of $D$, or (iii) $B$ was reached via an oceanlink.

3. Otherwise, if $B$ has a sibling and the sibling's water volume is less than its depression volume, then start from Step 1 with the new $B$ set as the depression pointed to by the current $B$'s geolink.

4. Otherwise, if $B$ has no sibling or the sibling's water volume is equal to its depression volume, then start from Step 1 with the new $B$ set as the parent of the current $B$ or, if $B$ has no parent, then use the depression to which $B$ oceanlinks.

The next step of the outer traversal, which begins one level in the DH closer to the ocean, identifies a less nested metadepression for which the inner traversal might need to be run. If this step were not supplied with information about prior water redistribution, it could cause the inner traversal to cover the same nodes repeatedly, which would be computationally wasteful. To prevent this, the inner traversal returns the ID of the final node in which it placed water: this node is the only node in the traversal with spare capacity so future traversals can begin there. Therefore, on subsequent overflows, if such a cached value is available, then the recursion skips directly to that node. This ensures that all the calls to this part of the algorithm take no more than $O(N)$ time collectively.

The following examples uses the geometry from Figure 2 to describe a set of inner traversals, starting with an overflowing Depression #12. Step numbers mirror those above; numbers in parentheses indicate the number of recursions – that is, the number of times that the inner-traversal algorithm has returned to Step 1:

    1 Depression #12 fills and overflows.

    2 Depression #12's water overflows into Depression #4, which is not full, following its geolink.

    1(r1) Depression #4 acts as Depression #12's parent via an oceanlink. The inner traversal terminates.

At this point, the outer traversal moves one level closer to the ocean, and the inner traversal repeats, this time starting at Depression #4.

    1 Depression #4 fills and overflows.

    2 Depression #4's water overflows into its sibling, Depression #5, which is not full and is a leaf depression. If Depression #5 had descendants, water overflowing from Depression #4 would have followed a geolink to one of these.

    1(r1) Depression #5s fills and overflows.

    2(r1) Depression #4 is full.

    3(r1) Depression #5 overflows into its parent, Depression #11.

    1(r2) Depression #11 overflows into the ocean; the inner traversal terminates.

Now the outer traversal moves yet another level closer to the ocean, and the new inner traversal starts at Depression #11.

    1 Depression #11 fills and overflows.

    2 Depression #11 has no sibling.

    3 Depression #11 overflows into its parent, the ocean; all remaining excess water is absorbed into an infinite sink.

    1(r1) The now-selected node is the ocean; the inner traversal terminates.

At this point, the outer traversal moves one level closer to the ocean, and arrives at the ocean. The outer traversal also terminates.

## 3.3 Flood the landscape

After water moves through the DH (Section 3.2, above), each node in the DH exists in one of the three following states:

1. **Empty:** The depression's water volume is equal to zero. In this case, nothing needs to be done. The depression's descendants might contain water, but the water never propagates to this level of the DH.

2. **Full:** The depression's water volume is equal to the volume of the depression itself. In this case, the depression is entirely full. If the depression's parent contains water, then the calculation of water depth is dealt with at a higher stage in the DH. If the depression's parent is empty, or if the depression's parent is the ocean, then the calculation is performed at this level as described below.

3. **Partially filled:** The depression's water volume is less than its depression volume. In this case, the depth of water across the depression and all its descendants' cells must be calculated at this level so that the depression fills to an appropriate level. This is described below and indicated as the *marginal volume* on Figure 2e.

The next step is to distribute this water across the DEM, appropriately flooding geographic depressions.

Given the three states described above, the algorithm locates the highest-level nodes which contain water. It does so via a post-order traversal. Each time the traversal reaches a leaf, the algorithm notes its label and pit cell. After identifying each of these, the algorithm reverses direction, moving from child to parent so long as the parent node contains water. Call the highest water-bearing node within a tree $L$.

In many cases, the water volume contained within the depression will be less than the total depression volume; therefore, we must calculate what the water level in the depression will be. To do this, we pick an arbitrary pit cell within $L$ and its descendants, and then use this as a seed from which to start building a priority queue which will traverse the cells of the depression. The priority queue returns cells ordered from lowest to highest elevation. At each step through the priority queue, the algorithm checks whether the cells visited so far collectively have enough volume to hold the water. If so, the algorithm exits, having successfully defined the flooded area. If not, it continues to use the priority queue to traverse the depression cell by cell. The filling procedure is shown in pseudocode in §6.3.

To expand this brief conceptual discussion into a more formal set of steps, let us begin by calling the active cell – that is, the one that is currently being considered by the algorithm – $c_p$. This cell is initially the arbitrary pit mentioned above, and is added to the priority queue. The algorithm marks $c_p$, which stands for "**c**ell of current highest **p**riority", as *visited*; all other cells remain unvisited. The algorithm then follows these steps:

1. Pop $c_p$ from the priority queue, call it $c$, and use its elevation to calculate the volume of water that can be accommodated in the set of cells processed so far (Equation 3, below). If this volume is enough to accommodate the volume of water available, exit the loop and compute the final water level (Equation 6, below). Otherwise, proceed to Step 2.

2. Add $c$ (which was popped in Step 1) to a plain queue, which records all of the cells scanned so far; these cells will later be inundated.

3. Add the cells neighboring $c$ that are not marked as *visited* to the priority queue if they belong to one of the descendant depressions of the one being filled. Each of these neighboring cells is then marked as *visited*.

4. Choose the lowest-elevation cell in the priority queue and label it as the new $c_p$ and return to Step 1. If the priority queue is empty, then all cells in the same meta-depression as $c_p$ or its descendants have been visited and we are now guaranteed to have sufficient depression volume to hold all of the water.

Step 1 in this approach requires an efficient way to determine the volume of a depression below any given elevation. If we call this elevation $z_o$ and the depression below the outlet contains $N$ cells with elevations $\{z_1, z_2, z_3, z_4, \ldots\}$ and unit cell area, the volume of water that the depression can accommodate simply equals the sum of the depth of water in each of its cells:

$$(z_o - z_1) + (z_o - z_2) + (z_o - z_3) + (z_o - z_4) + \ldots = No - z_1 - z_2 - z_3 - z_4 - \ldots \tag{1}$$

$$= No - \sum_{i=1}^{N} z_i \tag{2}$$

Now, consider cells $c_i = c_1, \ldots, c_N$ in the plain queue; that is, those cells that have been visited and popped from the priority queue. We can calculate the volume of water that can be accommodated in the depression below the elevation $z_s$ of the last cell $c_N$ (the sill) as:

$$V_{dep,z_s} = z_s \sum_{i=1}^{N} a_i - \sum_{i=1}^{N} z_i a_i \tag{3}$$

where $z_i$ is the elevation of cell $c_i$ and $a_i$ is the area of cell $c_i$. Thus, if we keep running sums while traversing the depression, it is possible to directly calculate the volume of water the depression can hold at each point in the traversal.

Once $V_{\mathrm{dep,z_s}}$ is greater than or equal to the volume of water in the depression, $V_w$, the plain queue contains all the cells to be flooded. At this point, the algorithm updates $z_w$, which is the water level within this depression. If $V_w = V_{\mathrm{dep,z_s}}$, the algorithm sets $z_w = z_N$. If instead $V_w < V_{\mathrm{dep,z_s}}$, the available volume in the depression is greater than the water volume, and the algorithm calculates $z_w$ in the depression as follows:

$$V_w = z_w \sum_{i=1}^{N} a_i - \sum_{i=1}^{N} z_i a_i \tag{4}$$

$$z_w \sum_{i=1}^{N} a_i = V_w + \sum_{i=1}^{N} z_i a_i \tag{5}$$

$$z_w = \left( \sum_{i=1}^{N} a_i \right)^{-1} \left( V_w + \sum_{i=1}^{N} z_i a_i \right) \tag{6}$$

We call Equation 6 the Lake-Level Equation (LLE). If all cells have a unit area, this simplifies to:

$$z_w = \frac{1}{N} \left( V_w + \sum_{i=1}^{N} z_i \right) \tag{7}$$

The conditional usage of the LLE described above is purely for computational efficiency: if $V_w = V_{\mathrm{dep,z_s}}$, its solution is that

$z_w = z_N$.

After solving for the water-surface elevation, the algorithm pops each cell in the plain queue ($c_i = c_1, \ldots, c_N$), corresponding

to the flooded region, and sets its water elevation to the computed $z_w$. This is the final step of the Fill-Spill-Merge algorithm. At

this point, it outputs a file representing the topography plus water thickness across the domain (i.e., topography with depressions

filled or partially filled with water).

Because Fill-Spill-Merge routes water cell-by-cell to the pit cells of depressions and manages an array of water depths, it

can be adapted for use with groundwater models, such as that described by Fan et al. (2013).

## 4   Algorithm performance

### 4.1   Theory

Here we use computational complexity as a means of contrasting the expected run-time of our algorithm against previous

algorithms such as FlowFill (Callaghan and Wickert, 2019). To do so, we describe a simple iterative solver similar to FlowFill

whose goal is to determine an appropriate water level for a depression. The solver operates on a one-dimensional domain of

cells bounded by high cliffs on either side in which each cell may have a column of water. At each step, if the solver finds a

discontinuity in water levels between two cells, it responds by averaging the heights of the cells' water columns. (The solver

we describe is known as Jacobi's method.) The challenge we present to this solver is a direct analogue of routing flow along a

stretch of river with negligible gradient and is very similar to routing flow across the surface of a lake or ocean.

For our analysis, we imagine that the system is initialized with a high column of water on the left and no water anywhere

else. We call the cell with the water Cell 1. We call the cells to its right 2, 3, 4, and so on. During the solver's first step, Cell 1

is initialized. On its second step, Cell 1 averages its height with Cell 2. On the third step, Cell 2 averages with Cell 3 and Cell

1 then averages with Cell 2. On the fourth step, Cell 3 averages to 4, 2 averages to 3, and 1 averages with 2. Thus, the number

of cells affected at each step are: 1, 2, 3, 4, and so on. Since there must be *at least* as many steps as there are cells, we can say

that there are $N$ steps. The total time, $t_{\mathrm{compute}}$, is then

$$t_{\mathrm{compute}} = \sum_{i=1}^{N} i = \frac{N(N+1)}{2} \tag{8}$$

Thus, for any model (Callaghan and Wickert, 2019; Fan et al., 2013) that uses a scheme similar to our simple solver, the time

required to solve the model is in $O(N^2)$.

In contrast, the new algorithm runs in $O(N \log N)$ time in the worst case. Moving water downhill (Section 3.1) is a flow-

accumulation algorithm. This is known to take $O(N)$ time (Mark, 1988) and efficient variants exist for performing flow

accumulation in parallel on large datasets (Barnes, 2017) and on GPUs (Barnes, 2019), though for simplicity we do not use

these techniques here. Moving water within the depression hierarchy (Section 3.2) requires a depth-first post-order traversal of

the entire hierarchy. This type of traversal is a foundational algorithm in computer science and takes $O(N)$ time. Each node

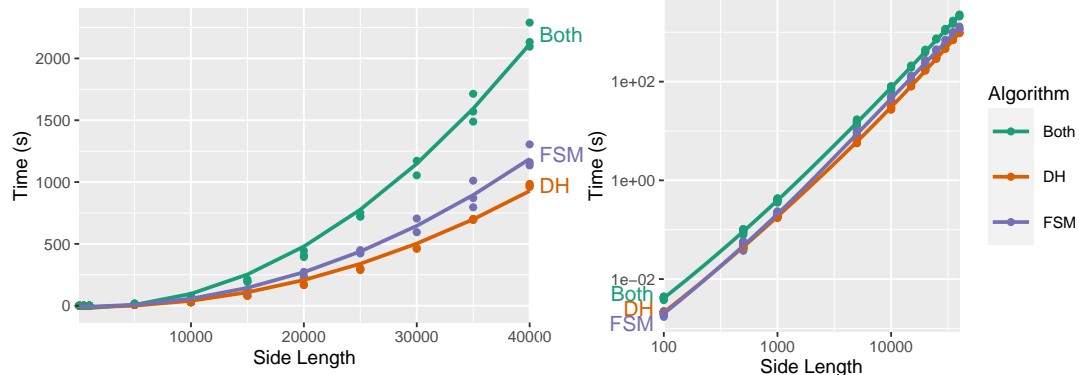

**Figure 6.** Performance on synthetic data. The left-hand plot shows the data on linear axes and the right-hand plot on log-log axes. The number of cells in each dataset is the square of the side length. The lines show $N \log N$ fits to each algorithm's time ($R^2 \approx 0.99$ for each). "DH" shows the performance of the Depression Hierarchy algorithm while "FSM" shows that of the Fill-Spill-Merge algorithm; "Both" shows the addition of these two values.

in this traversal has the potential to overflow, which also results in a depth-first traversal, thereby requiring up to $O(N)$ time.
However, by using a jump table that persists between calls to the overflow function, we ensure that it is able to identify the
target of the overflow in amortized constant time; that is, the function is able to skip over fully-filled depressions. Finally, the
algorithm floods the digital elevation model from the pit cells up. This requires a depth-first post-order traversal, which calls
a flooding function (Section 3.3) on select subtrees of the DH. The depth-first traversal takes $O(N)$ time, as described above.
The priority queue used for flooding nominally takes $O(N \log N)$ time in the worst case for floating-point data and $O(N)$
time in the worst case for integer data (Barnes et al., 2014). However, with specialized data structures the time can be reduced
to $O(N)$ for both floating-point and integer data (Barnes et al., 2014). Most real datasets consist of many small depressions
whose cell counts $N_{\text{cells-in-dep}}$ are much smaller than the total number of cells in the digital elevation model. Therefore, the
actual time is for this step is $O(N_{\text{dep}} N_{\text{cells-in-dep}})$, where $N_{\text{dep}}$ is the total number of depressions and $N_{\text{dep}} N_{\text{cells-in-dep}}$ can
be much less than $N$. Because the worst-case time complexity of any operation is $O(N)$, this bounds the time of the algorithm
as a whole. However, to reduce the potential for bugs, we use the C++ standard library's $O(N \log N)$ priority queue in our
implementation, at the cost of reduced performance.
To put this in more concrete terms, consider a long stretch of low-gradient river. Such a feature poses a lower bound on the
time of our simple solver. North America's Red River of the North runs for 885 km with a gradient that is often on the order of
$0.03 \, \text{m} \, \text{km}^{-1}$. On a 90 m grid of floating-point data, the river would be 9,833 cells long. Our simple (Jacobi) solver would then
take an estimated 97 million time units to reach a solution, whereas the new solver that we describe in this paper would take
9,833 time units, a 10,000$\times$ speed-up. Our empirical results, below, support both the theory and this expected value.

| Dataset | Dimensions | Cells | FSM Time [s] | Total Time [s] |
|---|---|---|---|---|
| Madagascar | 2000×1000 | $2.0 \cdot 10^6$ | 0.1 | 0.4 |
| U.S. Great Basin | 1920×2400 | $4.6 \cdot 10^6$ | 0.2 | 8.7 |
| Australia | 5640×4200 | $2.3 \cdot 10^7$ | 9.1 | 15.6 |
| Africa | 9480×9000 | $8.5 \cdot 10^7$ | 65.3 | 118.0 |
| N&S America | 18720×17400 | $3.2 \cdot 10^8$ | 53.2 | 231.6 |
| Minnesota 30m topobathy | 34742×23831 | $8.2 \cdot 10^8$ | 307.8 | 792.6 |

**Table 1. Datasets used, their dimensions, and algorithm wall-times.** Tests were performed on the Comet cluster run by XSEDE (see main text for full specifications). Times for Fill-Spill-Merge ("FSM Time") alone and this time plus the depression hierarchy construction time ("Total Time") are shown. Topographic data for Madagascar, the U.S. Great Basin, Australia, Africa, and North & South America, were clipped from the global GEBCO_08 30-arcsecond global combined topographic and bathymetric elevation data set (GEBCO, 2010). The Minnesota 30m topobathy data is the merged result of two data sources. The topography is resampled from the Minnesota Geospatial Information Office's 1m LiDAR Elevation Dataset (MNGEO - Minnesota Geospatial Information Office, 2019). Bathymetric data were provided by the Minnesota Department of Natural Resources (MNDNR - Minnesota Department of Natural Resources, 2014). Richard Lively of the Minnesota Geological Survey merged and combined these data sets.

## 4.2 Computational Performance

We have implemented the algorithm described above in `C++17` using the Geospatial Data Abstraction Library (GDAL) (GDAL Development Team, 2016) to read and write data. There are 924 lines of code of which 50% are or contain comments. The code can be acquired from https://github.com/r-barnes/Barnes2020-FillSpillMerge and Zenodo (Barnes and Callaghan, 2020). The code contains extensive unit and end-to-end tests, which leverage both deterministic and random testing; the code passes a total of 214,990 test assertions and achieve 97% test coverage. The missed lines flag emergency situations which can only arise if there is a logic error, so they (in theory) cannot be reached.

Tests were run on the Comet machine of the Extreme Science and Engineering Discovery Environment (XSEDE) (Towns et al., 2014). Each node of the machine has 2.5 GHz Intel Xeon E5-2680v3 processors with 24 cores per node and 128 GB of DDR4 DRAM. Code was compiled using GNU g++ 7.2.0 with full optimizations enabled.

We ran two sets of scaling tests, one on actual data and one on synthetic data. On actual data, our scaling tests cover datasets spanning three orders of magnitude in terms of their number of cells, as shown in Table 1. The R package GuessCompx Agenis-Nevers et al. (2019) shows that an $O(N \log N)$ scaling relationship gives the best fit to the data, which agrees with the theory.

To more precisely demonstrate performance, we run Fill-Spill-Merge on synthetic landscapes of various sizes generated using RichDEM's Perlin noise random terrain generator (Barnes, 2018). Multiple landscapes are generated and timed at each size to smooth timing variation due to both the data and fluctuations in the testing environment. This results in Figure 6, which again shows that the performance data gives a good fit to an $N \log N$ function.

## 4.3 Model intercomparison

Given a depression hierarchy data structure, Fill-Spill-Merge provides an efficient method to route water across any surface while taking depressions into account. Furthermore, Fill-Spill-Merge can be used to assess which depressions are most important in day-to-day or seasonal changes to the hydrologic system. For example, small depressions will become flooded and spill over even with relatively small amounts of water reaching them, while larger depressions may not be completely filled. These depressions impact the hydrologic connectivity of the landscape (Callaghan and Wickert, 2019). If standing water is retained between invocations of Fill-Spill-Merge, and new water added at each invocation, the algorithm can be used to simulate the movement of water across landscapes; we will explore this further in future work.

We have compared Fill-Spill-Merge with a prior algorithm, FlowFill, at the same two sites used by Callaghan and Wickert (2019): a reach of the Sangamon River in Illinois (Figure 7) and the Río Toro basin in Argentina (Figure 8). Like Fill-Spill-Merge, FlowFill can be used to route water across a landscape while preserving real depressions, but the latter algorithm is significantly slower (Table 2). The two selected study sites provide very different landscapes for testing the performance of the algorithm. The Sangamon River site is located at 39.97°N, 88.72°W, in Illinois, USA. It is a low-relief, post-glacial landscape containing many closed depressions, which may impact hydrologic connectivity as a function of runoff (Lai and Anders, 2018). It furthermore contains a grid of roads and associated embankments whose elevations are significant when compared to regional relief and impact water flow paths and storage. Callaghan and Wickert (2019) resampled the 2.5 ft (0.76 m) resolution LiDAR DEM (Illinois Geospatial Data Clearinghouse, 2020) to 15 m resolution for analysis and manually removed several road bridges using GRASS GIS (Neteler et al., 2012) to prevent artificial pooling behind these; here we use the same modified DEM to enable a direct comparison between the algorithms. The Río Toro site is located mainly in Salta Province, Argentina, around 24.5°S, 65.8°W. This site exhibits more rugged fluvially sculpted topography (Hilley and Strecker, 2005). Callaghan and Wickert (2019) resampled the 12-m TanDEM-X DEM of this region (Krieger et al., 2013; Rizzoli et al., 2017) to 120 m resolution. Here we use this same resampled DEM for comparison. The runoff depths used at each of the two study sites were selected to show a range of water levels present in the depressions. The depths shown were therefore scaled based on the amount of water required to completely fill depressions in the landscape.

As shown in Table 2, wall-times using Fill-Spill-Merge ranged from 0.227–0.243 s for the Sangamon River site and 0.300–0.319 s for the Río Toro site. This compares with times ranging from 20–643 s and 31-155 s, respectively, for FlowFill. These times for both sites correspond to a 86–2,645× reduction in wall-time using FSM. Since FlowFill was run with 24 processors, this translates to a 2,064–63,480× reduction in compute time. Considering that each of these example DEMs is quite small relative to modern full-resolution LiDAR-derived elevation data sets or continental-scale 30-meter DEMs (Table 1), this speed-up and its associated $O(N \log N)$ scaling provides a significant advantage for topographic analysis and solving associated problems in hydrology and geomorphology.

Although both FlowFill and Fill-Spill-Merge route water downslope, flooding depressions based on the quantity of available water, our FSM results differ in some ways from those of FlowFill (Callaghan and Wickert, 2019). In both Figures 7 and 8, Fill-Spill-Merge flooded some depressions more deeply than FlowFill did and flooded some depressions with less water. At

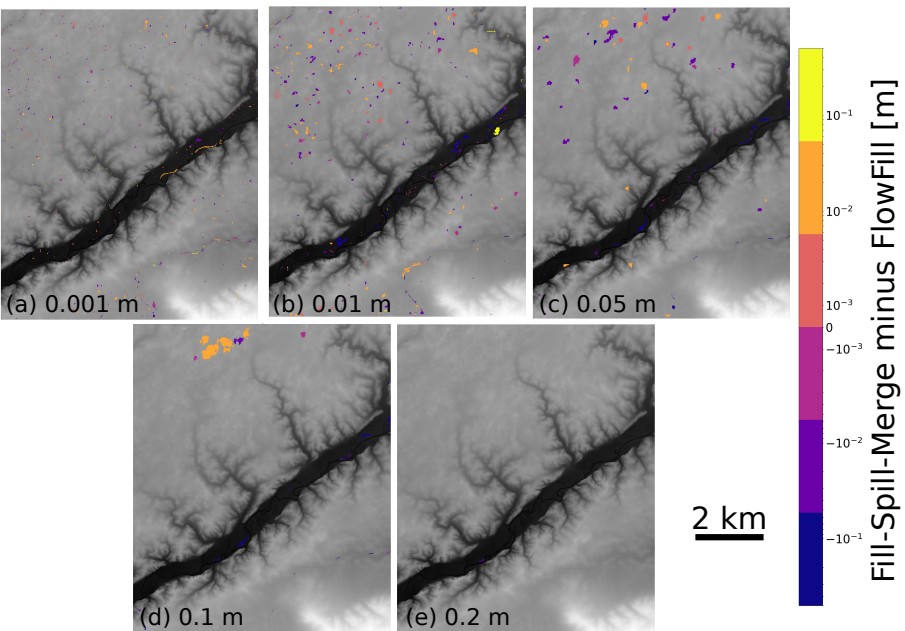

**Figure 7.** The difference between results of Fill-Spill-Merge and FlowFill at the Sangamon River site. The values for panels (a) to (e) indicate the depth of uniform runoff applied across the landscape for both algorithms. For example, in (a), each cell across the domain starts with 0.001 m of surface water. Orange to yellow colors indicate locations where Fill-Spill-Merge had more water, and purple to blue colors indicate locations where FlowFill had more water. Differences of less than 3 mm have been masked out. Differences are generally small, and are likely a result of the iterative nature of the FlowFill algorithm which causes it to asymptotically approach the correct values. In some locations, Fill-Spill-Merge retains slightly more water in depressions that FlowFill does. This could be due to water which has not yet finished moving downslope and into these depressions in the FlowFill algorithm. In other locations, FlowFill has retained more water. One possible reason for this is that some depressions have a narrow outlet, through which Fill-Spill-Merge is able to move all water as appropriate but the cell-by-cell movement of water with FlowFill can produce transient dams that reroute additional water towards these subcatchments. This DEM was prepared by Lai and Anders (2018) and Callaghan and Wickert (2019) from LiDAR topographic data provided by the Illinois State Geological Survey (Illinois Geospatial Data Clearinghouse, 2020).

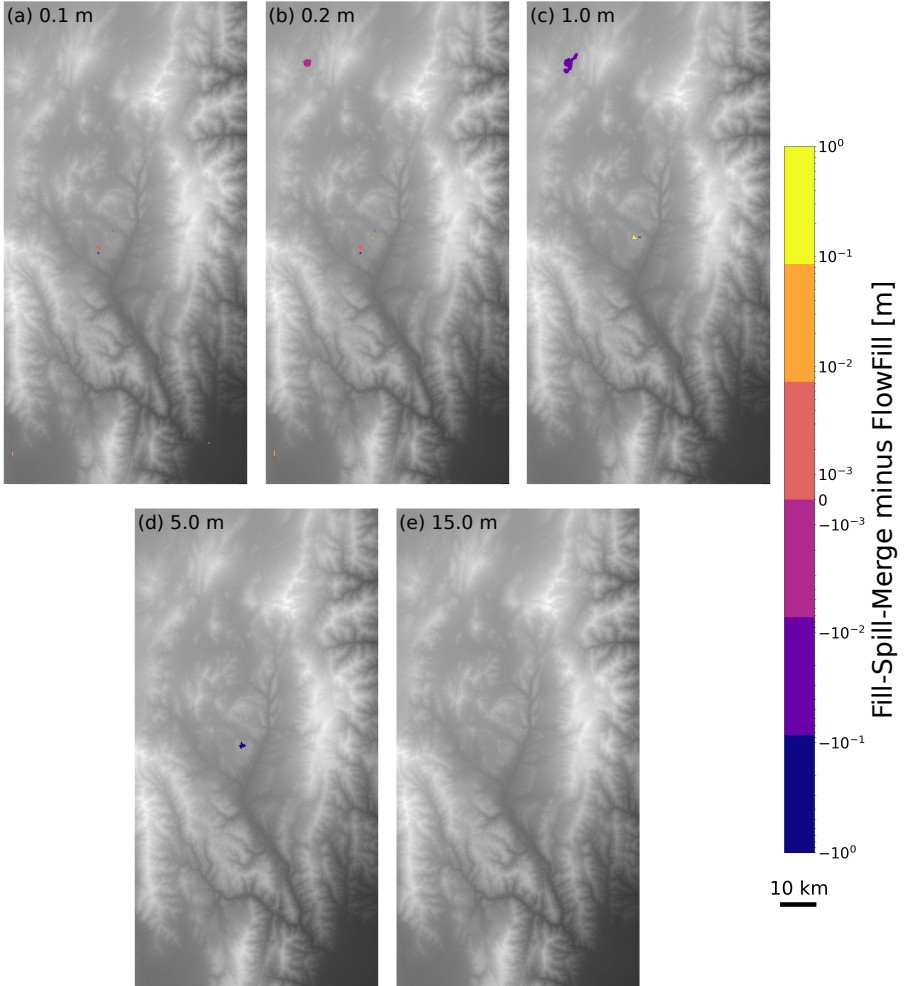

**Figure 8.** The difference between results of Fill-Spill-Merge and FlowFill at the Río Toro site. The values for panels (a) to (e) indicate the depth of uniform runoff applied across the landscape for both algorithms. For example, in (a), each cell across the domain starts with 0.1 m of surface water. Orange to yellow colors indicate locations where Fill-Spill-Merge had more water, and purple to blue colors indicate locations where FlowFill had more water. Differences of less than 3 mm have been masked out. In panel (e), 15 m of water was enough to fill all depressions with both algorithms, so there are no differences between the two. The most significant difference is seen in panel (c), where FlowFill retained additional water in a large depression. This is likely due to transient damming of its narrow inlet in FlowFill's cell-by-cell method of moving water, which may have prevented the full volume of water from leaving the depression. This DEM was generated with data acquired from the TanDEM-X mission (Krieger et al., 2013; Rizzoli et al., 2017).

| | Sangamon | | | Río Toro | | |
| Runoff depth [m] | FlowFill [s] | FSM [s] | Speed-up [x] | FlowFill [s] | FSM [s] | Speed-Up [x] |
|---|---|---|---|---|---|---|
| 15 | 642.65 | 0.243 | 2645 | 154.70 | 0.317 | 488 |
| 10 | 626.59 | 0.241 | 2600 | 124.37 | 0.309 | 402 |
| 5 | 570.02 | 0.241 | 2365 | 93.56 | 0.300 | 312 |
| 1 | 472.33 | 0.241 | 1960 | 53.09 | 0.316 | 168 |
| 0.2 | 508.87 | 0.235 | 2165 | 38.30 | 0.316 | 121 |
| 0.1 | 464.15 | 0.230 | 2018 | 35.75 | 0.301 | 119 |
| 0.05 | 418.71 | 0.243 | 1723 | 33.62 | 0.316 | 106 |
| 0.01 | 200.81 | 0.227 | 885 | 32.06 | 0.315 | 102 |
| 0.001 | 20.12 | 0.235 | 86 | 30.99 | 0.319 | 97 |

**Table 2. Time comparison of Fill-Spill-Merge vs FlowFill.** Wall-times are in seconds comparing FlowFill (Callaghan and Wickert, 2019) parallelized across 24 cores versus Fill-Spill-Merge on a single core; "Speed-Up" is a multiplicative factor. Using FlowFill, wall-times increased with the depth of applied runoff and on flatter landscapes. Using FSM, wall-time is independent of depth of applied runoff and ruggedness of landscape, but increases for larger domains. FSM's wall-times were 86–2,645 times faster than FlowFill for these examples; compute times were 2,064–63,480 times faster.

both study sites, the differences between the two algorithms are minimized at the extreme high and extreme low starting runoff values. For the highest runoff values, this is because both algorithms successfully fill all depressions in the landscape, so that no differences are possible. For the lowest runoff values, both algorithms simulate only a small amount of water filling any depression, so that that significant differences between the two algorithms are not possible. The biggest differences are therefore seen for moderate starting runoff values, when depressions contain substantial water volumes, but are still only partially filled. One possible cause for these discrepancies is FlowFill's asymptotic approach to an equilibrium water level, which may prevent small volumes of water from reaching the depression to which they belong. On the other hand, depressions with a narrow outlet could be especially prone to being overfilled by FlowFill because its cell-by-cell algorithm could dynamically dam this outlet, routing additional water into the depression. Both of these possibilities are further linked to the fact that FlowFill dynamically evolves a land-plus-water flow-routing surface, whereas Fill-Spill-Merge routes flow just over the land surface. These differences make FlowFill more useful for understanding temporal changes in surface water distribution, while Fill-Spill-Merge provides a more accurate snapshot of surface hydrology under equilibrium conditions.

## 5 Conclusions

Here we leverage the depression hierarchy data structure (Barnes et al., 2020) to route flow through surface depressions in a realistic, yet efficient, manner. In comparison to previous approaches, such as Jacobi iteration, the new algorithm runs in log-linear time in the input size and is accompanied by extensively commented source code. This computationally efficient al-

gorithm may help us to better understand hydrologic connectivity and water storage across the land surface, and is an important step forwards in recognising the importance of depressions as real-world features in digital elevation models.

*Code availability.* Complete, well-commented source code, an associated makefile, and correctness tests are available from https://github. com/r-barnes/Barnes2020-FillSpillMerge and Zenodo (Barnes and Callaghan, 2020).

*Author contributions.* KC and AW conceived the problem. RB conceived the algorithm and developed initial implementations. KC and RB completed, debugged and tested the algorithm. All authors contributed to the preparation of the manuscript.

*Competing interests.* The authors declare that they have no conflict of interest.

*Acknowledgements.* RB was supported by the Department of Energy's Computational Science Graduate Fellowship (Grant No. DE-FG02-97ER25308) and, through the Berkeley Institute for Data Science's PhD Fellowship, by the Gordon and Betty Moore Foundation (Grant GBMF3834) and by the Alfred P. Sloan Foundation (Grant 2013-10-27).

KLC was supported by the National Science Foundation under grant no. EAR-1903606, the University of Minnesota Department of Earth Sciences Junior F Hayden Fellowship, the University of Minnesota Department of Earth Sciences H.E. Wright Footsteps Award, and start-up funds awarded to AW by the University of Minnesota.

Empirical tests and results were performed on XSEDE's Comet supercomputer (Towns et al., 2014), which is supported by the National Science Foundation (Grant No. ACI-1053575). Portability and debugging tests were performed on the Mesabi machine at the Minnesota Supercomputing Institute (MSI) at the University of Minnesota (http://www.msi.umn.edu).

The Deutsches Zentrum für Luft- und Raumfahrt (DLR) provided 12 m TanDEM-X DEM coverage of the Río Toro catchment via proposal DEM_GEOL1915 awarded to Taylor Schildgen, Andrew Wickert, Stefanie Tofelde, and Mitch D'Arcy. Jingtao Lai and Alison Anders provided a copy of their Sangamon River DEM.

This collaboration resulted from a serendipitous meeting at the Community Surface Dynamics Modeling System (CSDMS) annual meeting, which RB had attended on a CSDMS travel grant.

# 6 Pseudocode

## 6.1 MoveWaterInDepHier

```
1: function MoveWaterInDepHier(root, DH, JumpTable)
2: Let root be the id of the depression we're currently con-
   sidering
3: Let DH be a Depression Hierarchy
4: Let JumpTable be a hash table mapping DH labels to DH
   labels
5:
6: ▷ For "children" of leaves
7: if root=NOVALUE then return
8:
9: ▷ The traversal
10: for each ocean-linked child c of root do
11:     Call MoveWaterInDepHier(c, DH, JumpTable)
12: end for
13: Call MoveWaterInDepHier(c.left_child, DH, JumpTable)
14: Call MoveWaterInDepHier(c.right_child, DH, JumpTable)

15:
16: if root=OCEAN then return
17:
18: if root has children and both their depression volumes
    equal their water volumes and root's water volume is zero
    then
19:     root.water_vol += root.left_child.water_vol
20:     root.water_vol += root.right_child.water_vol
21: end if
22:
23: if root.water_vol>root.dep_vol then
24:     Call OverflowInto(root, root.parent, DH, JumpTable, 0)
25: end if
```

## 6.2 OverflowInto

```
1: function OverflowInto(root, StopNode, DH, JumpTable,
   ExtraWater)
2: Let root be the id of the depression we're currently con-
   sidering
3: Let StopNode be the id of the depression that ends the
   traversal. It is the parent of the depression that first called
   this function.
4: Let DH be a Depression Hierarchy
5: Let JumpTable be a hash table mapping DH labels to DH
   labels
6: Let ExtraWater be the water that needs to be distributed
   in DH
7:
8: ▷ If depression is too full, get its excess so we can find a
   home for it
9: if root.water_vol>root.dep_vol then
10:     ExtraWater += root.water_vol - root.dep_vol
11:     root.water_vol = root.dep_vol
12: end if
13:
14: if root=StopNode or root=OCEAN then
15:     root.water_vol += ExtraWater
16:     return root
17: end if
18:
19: ▷ 1st place to stash water: in this depression
20: if root.water_vol<root.dep_vol then
21:     Let C=root.dep_vol - root.water_vol
22:     if ExtraWater< C then
23:         root.water_vol = root.water_vol+ExtraWater
24:         ExtraWater = 0
25:     else
26:         root.water_vol = root.dep_vol
```

27:     *ExtraWater -= C*

28:   **end if**

29: **end if**

30:

31: **if** *ExtraWater=0* **then**

32:   **return** root

33: **end if**

34:

35: **if** *root∈JumpTable* **then**

36:   **return** *JumpTable*(root) = OverflowInto(*JumpTable(root), StopNode, DH, JumpTable, ExtraWater*)

37: **end if**

38:

39: ▷ 2nd place to stash water: in the depression's sibling

40: **if** *root.sib≠*NOVALUE **then**

41:   **if** *root.sib.water_vol<root.sib.dep_vol* **then**

42:     **return** *JumpTable(root)* = OverflowInto(*root.geolink, StopNode, DH, JumpTable, ExtraWater*)

43:   **else if** *root.sib.water_vol>root.sib.dep_vol* **then**

44:     *e=root.sib.water_vol-root.sib.dep_vol*

45:     *ExtraWater += e*

46:     *root.sib.water_vol = root.sib.dep_vol*

47:   **end if**

48: **end if**

49:

50: ▷ 3rd place to stash water: in the depression's parent

51: **if** *root.parent.water_vol=0* **and** *root* is not oceanlinked to *root.parent* **then**

52:   *root.parent.water_vol += root.water_vol*

53:   **if** root.sib≠NOVALUE **then**

54:     *root.parent.water_vol += root.sib.water_vol*

55:   **end if**

56: **end if**

57: **return** *JumpTable(root)* = OverflowInto(*root.parent, StopNode, DH, JumpTable, ExtraWater*)

## 6.3   FillDepressions

1: **function** FillDepressions(*PitCell, OutCell, DepLabels, WaterVol, dem, labels, wtd*)

2: Let *PitCell* be the cell to start filling from

3: Let *OutCell* be the outlet/spill cell

4: Let *DepLabels* be the labels contained within the metadepression we are trying to fill

5: Let *WaterVol* be the amount of water that needs to be spread throughout the depression

6: Let *dem* be the topography.

7: Let *labels* be the labels from the Depression Hierarchy

8: Let *wtd* be the depth of water in each cell.

9: Let *visited* be a hash set of cell ids

10: Let *PQ* be a priority queue sorted by increasing elevation

11: Let *affected* be a plain queue

12: Let $T_e$ be the total elevation; initially 0

13:

14: **if** *WaterVol=0* **then return**

15:

16: Place *PitCell* into *PQ* and mark it visited

17: **while** *PQ* is not empty **do**

18:   Let *c=pop(PQ)*

19:   Let $V = |affected| \cdot c.elev - T_e$

20:

21:   **if** *WaterVol< V* **then**

22:     $W_L = (WaterVol + T_e)/|affected|$

23:     Set *wtd* for all cells in *affected* to $W_L$

24:     **return**

25:   **end if**

26:

27:   **if** $c \neq OutCell$ **then**

28:     Place *c* into *affected*

29:     $T_e$ += *c.elev*

30:   **end if**

31:     Add all of $c$'s neighbours that belong to depressions in
*DepLabels* and are not the outlet cell to *PQ* and mark
them visited
32:  **if** *PQ* is empty **then**
33:     Add *OutCell* to *PQ* and mark it visited
34:  **end if**
35: **end while**

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
