# Peer review of "Computing water flow through complex landscapes – Part 3: Fill-Spill-Merge: Flow routing in depression hierarchies"

_Earth Surface Dynamics, 2020_

## Referee Comment (RC1) · Anonymous Referee #1 · 26 May 2020

In this paper, Barnes et al present the details of the Flow-Spill-Merge algorithm (FSM) already introduced in prior papers. I am not an expert in C++ so I can't comment on the code. I think the paper is well-written and needs only minor adjustments:

figure 1: - I suggest to number the depressions in the figure to make it easier for the reader to follow the explanations

lines 70-83: I think that if the authors refer to the subfigures it will be easier to read. I found that fig.2e has the elements I needed to understand the text.

line 78: when #0 in mentioned in fig.2, I had a bit of trouble finding it in the figure (it's a bit hidden under "Ocean", and the purple color of the square in 2a is a bit dark)

[Figure]

line 93: figure 2e

line 100: figure 2a

line 173: the I was a bit confused by the "water than they can hold spill" here. Is it "water than they can hold will spill"?

line 382: maybe insert a citation for GRASS GIS here? (Neteler et al 2012)

figures 6 and 7: Have you thought about using a divergent colorscale here? Since you are showing positive and negative differences?

- Is the drainage network from both algorithms compared here are identical, differing only at the depressions? I was curious to see the drainage.

---

## Referee Comment (RC2) · Daniel Hobley (Referee) · 29 May 2020

This paper presents an efficient, novel method for the routing of flow across depressions while still honouring the mass balance of the water flux. This is an important computational geomorphic problem of long-standing, and this submission is certainly worthy of publication in ESurf. I would expect the algorithm to be taken up widely by the surface process and hydrological modelling communities.

I should note that I have not reviewed parts 1 and 2 in any great detail; this review focuses specifically on this as part 3. I will leave it to someone higher up the editorial chain to ensure there isn't too much overlap here, but based just on this manuscript, it

doesn't seem like there will be.

Thanks to the install issues I've had (see technical comments), I haven't been able to actually run the software. However, once installed cleanly, I imagine this will work fine. Happy to quickly confirm after corrections have been made on the manuscript – but thought you'd rather not have the delay on getting these comments while we sort annoying OS and versioning issues on Github.

I like this manuscript a lot, and can recommend it be published with what I think are minor and hopefully pain-free revisions. I think that the only comments I have that would involve substantial changes to the work are around a. being more clear in describing the algorithm, and b. adding a few synthetic performance tests.

I have provided some general comments, some line item comments, and some technical comments (i.e., on the code itself) below.

I look forward to seeing this in print in short order!

Dan Hobley

—

General comments on the article

In general, this is an extremely well-written manuscript, especially considering the audience likely to see this paper in ESurf and the level of technicality that is clearly actually involved here under the hood. I've pulled the following broader comments out because they don't have specific linenumbers, rather than because they are particularly major.

\* I found the way the depression hierarchy is talked about in the Algorithm section quite confusing. I think the reason is that the authors convolve the object that is the hierarchy itself (i.e., the input to this algorithm), the conceptual idea of the DH, and the method used to build that hierarchy (as explained in Barnes et al. (2020)). For example around P5ln30 "The specific outputs from the depression hierarchy [the method] that

are used in the Fill-Spill-Merge algorithm are: – DH: the depression hierarchy [the object] itself….”; P8ln2 “By routing water according to the DH…” -> so is this the object, or the method? Finding a distinct way to refer to the method/algorithm that produces a DH, the ordering that is described by a DH, and a DH structure itself would significantly enhance clarity throughout. (See line item P9ln26 also.)

* Jargon suddenly appears at the end of p.9. The whole section P9ln25-P10ln17 I found very confusing. I ramble about this at length in the line comments below, but at heart this needs a. the jargon explaining, b. the text simplifying, and c. the text expanding as necessary.

* Regardless of that, I would strongly advocate that you put some actual pseudocode into the paper – maybe as supplemental material, but in the body would also be good. This would really help in sections like that one, and the one to follow (both noted below).

* Similarly, if iterations like those on P10 are described, then I think they would be much clearer with cartoons illustrating the possible steps next to them. You use fig 2 as a case study for the sequence on P11, but it would be much clearer to produce a second version along the lines of fig 2 but with multiple parts. These would highlight each inner loop as described in the text, what happens under the possible configurations of that loop, and the sequencing of the water filling in under the rules for those configurations.

* Oceanlink-geolink terminology. I like the conceptual framework of the paper a lot, but the description of “paired oceanlink-geolinks” I found confusing. Because there can never be an oceanlink without a geolink, I would tweak the terminology such that an oceanlink is a subset of a geolink, and the pair concept goes away.

* The maths as described on P13 is not strictly correct. The equations give summed elevations, but they are variously described as “capacity” or “volume”. There should be cell areas in here somewhere. Also, there is a missing definition of N. Check the maths again for precision of terms.

* Section 5 – The text asserts that O(NlogN) is the best fit, but the paper does not show it. At the very least, plot the data in Table 1. . . .however, doing this will reveal that the data points do not in fact give a good smooth correlation, convincing of NlogN (which is presumably why you didn't do it in the first place!) I would retain Table 1 as nice examples of "real world" model performance, but add a synthetic test with random noise elevation on varying size grids. This should give you the convincing NlogN scaling to plot up and push a NlogN scaling best fit through, and is also a cleaner – and more standard - way making the point about actual performance of your implementation. Some light restructuring (see below) would help here too.

* You could add a statement somewhere that this algorithm also permits water to be retained in the depressions between runs, if that is desired – so you can get a timeseries as the lakes fill over time. It would also easily let you perform true water balances on the lakes if the user wanted to, by subtracting water between steps. This is implicit in the text, but it's a cool feature you could emphasise a bit more if you want.

—

Line-item comments:

P1ln16 – misformatted ref

P3ln15 – 0 seems a special case here. Refer also to 12 or 11?

P5ln10 – The ocean links. You should make it clear here why, from a modelling perspective, you need to separate ocean links from geolinks, so we know where we're going. Also, you have inconsistency between "ocean link" and "oceanlink" – check for consistency (incl. In figure captions).

Fig 4. This caption could be a lot more generous to the reader. Explain the colorcoding; what *exactly* the blue shading is; and exactly what you mean by "non-additive". I don't quite follow, and feel there must be a clearer way of expressing this. Also, would it not be clearer to actually let area represent volume of water? (I assume that this is not the

case is what you are getting at by "non-additive"?) In particular g-j are quite hard to follow and need more text. Isn't j inconsistent with f? You appear to suggest in f the whole system floods out, but j does not show this. Format-wise, you have a,b,c on the fig and A,B,C in the text.

P9ln10 – "depth/breadth-first traversal". Please explain what this is in plainer English.

P9ln25 – Suddenly, the paper becomes much more jargon-y and hard to follow. Clarifying depth-first at ln 10 will help, but what is "post-order"? "an invariant"? Please explain these in plainer English. This paragraph and the next are also quite confusing aside from the jargon, so please expand and be more generous with the text as well. Things I am struggling with: How does a depth-first traversal have an "outer" rather than a "deeper"? Clearly the water starts in the leaves, so how can the outer (first??) traversal begin in the ocean (ln 30), which is a trunk? How are the traversals nested, and when do we shift from outer to inner and vice versa? An explanation of the jargon in a pre-paragraph will probably sort most of this – i.e., what are outer and inner traversals, what is pre- and post-order, and how are these concepts interrelated? Another flow-chart, a diagram, or chunk of pseudocode would also really help here. The steps on P10 help piece together what these paragraphs must mean, but they come after the confusing text not with it, and the text should really stand on its own.

P10ln10 – "We add water *from A* to B"? Where does the added water come from? Is it the original volume or the passed volume? (Must be the latter, but this is not as written) Tweak the phrasing here.

P10ln11 – "this part of the algorithm". Do you mean point 1, the inner traversal, or the section 3.2? Point (iii) in this list is also very confusing. Break this case out and say what you mean (and why this is a terminating case) more clearly. The manuscript has not described the concept of an oceanlink-geolink pair (surely the link between two nodes is either an oceanlink or a geolink, so there can be no pairing?), so clarify that too (see main comment). Again, this feels like a diagram or pseudocode or both would

add a lot here. The diagram I have in mind would show the possible cases for this 1-3 loop as a sequence of filling lakes, as in Fig 2 or fig 4a,b, g-j (but where water volume is honoured). Although I appreciate this section looks like this because it explicitly mirrors the structure of the code, there is surely a clearer way to express 1-3. E.g.

– 1. If a node has excess water, attempt to pass that water to an unfilled sibling.

– 2. If no unfilled sibling is available, pass the water to the parent and terminate inner loop.

– 3. If no parent exists, the node must have an oceanlink, so pass the water along the oceanlink and terminate inner loop.

– 4. If the oceanlink ends in the ocean, discard the excess water and terminate inner loop.

Does this not encompass all the logic without the detail in the subcases, the pseudo-parents, etc.?

P10ln31 – apart from still disliking the "paired" terminology here, this also brings back the pt (iii) I didn't understand above. This makes it clear what you meant there, but the idea that 12 is sort of a parent of 4, but not really, and only sometimes, is a confusing one. I would stick with the idea that parents are strictly in the same tree, and replace this idea with simply "passing water down the oceanlink to move it between trees". I don't see what else it is adding for the cost of being confusing. In general though, this stuff on P11 is really good.

P12lns3-4 – I don't see why the algorithm needs to start at the ocean and iterate back to every leaf, only to iterate forward again. Given we already maintain a list of all the leaves, associated pits, and labels, why can't me just start at the leaves and only go forward? If I'm missing something, it means this needs clarifying anyway. Are you trying to say, iterate from the ocean down along all the branches, but pinch out the branch whenever you find water and don't go further?

P12lns15-28: Another place where a block of pseudocode alongside the text would add a lot. Arguably it could replace the text entirely, but having both would be extra clear.

P12ln29. "To do so, we imagine a hypothetical outlet that drains the depression... add:"at this elevation". This section for me says the same thing three times in different words. I would have simply: "Step 1 in this approach requires an efficient way to determine the volume of a depression below any given elevation. To do so, we imagine a hypothetical outlet that drains the depression at that elevation. If we call the elevation of this hypothetical outlet o and..." (also see comment below about maybe removing the idea of a hypothetical outlet entirely.)

Equation 1: You don't actually define N

Equation 3: I don't object to it, but why is there a bar through V?

P13ln4 – Run-on; end with a period not comma or tweak phrasing.

P13ln5 – "the sill". Which? The hypothetical one, or the real one?

In general here, I think the concept of an "imaginary" outlet at every possible node height is getting in the way of what you're trying to say. Isn't it just that you can track the volume of a depression below various depths using the equations you give by cumulative summing along the plain queue, and then can compare that increasing volume as you move along the queue to the water volume available, which tells you the fill level? I would rephrase the way you present P12ln29-P13ln9 without this imaginary sill/outlet concept.

P13ln21 – I don't think this is the right heading for the section. Isn't this "Algorithm performance" or something like that more specific? 4 and 5 would then be subsections, e.g., "theory", "computational performance". The section itself reads really well though – and I assume by this point you will have defined the iteration jargon a bit better at first use.

P14ln32 – If you rename the section, I don't think this final sentence is needed.

P15ln11 – consider "Applications and testing on real-world data" or words to that effect (assuming you renamed 4 above, and remove the subheading below).

P16ln1 – there is no 6.2!? Remove the subheading. Otherwise, I like this section a lot.

...overall, I think a light restructuring (i.e., broadly just some section renaming and text shunting) through 4-6 would make this a bit stronger. What you have is testing of performance on both synthetic (please!) and real-world data, then a real-world case-study that attempts some model validation. I think the sectioning should reflect that a bit more explicitly, e.g. something like,

4. Model performance

– 4.1 Theoretical analysis

– 4.2 Performance on synthetic data

5. Field testing and intermodel comparison

– 5.1 Performance on real world data

(brief chat about Table 1 to give people an idea of general performance on real-world data, or just remove and junk these subsections)

– 5.2 Model intercomparison

(This is basically just 6.1 – and by all means, keep the performance data for this case study within this section)

Conclusions do the job well.

—

Technical comments

Install instructions in the readme require cmake – and so should probably explicitly

direct the user to acquire this first. I note configuration issues are also likely, which should probably also be clarified (see below); if they are not, the authors are dramatically reducing the potential reach and impact of their work. In general, I think the readme needs expanding a bit (see technical comments).

I severely struggled with the install for the software, and am probably significantly more tech-savvy than many target users. Did the authors test installs thoroughly against a full range of OSes? I am running macOS Catalina, XCode 11.4 (i.e., fully modern installs) and anaconda, and am seeing a spew of cmake errors which some investigation reveals are related to incompatibilities between the installer, the shipped SDKs with XCode, and anaconda. This is going to be a common user case, and I think the authors need to address it, if only briefly, in the docs – and test some other common environments too for safety's sake.

[Part of the answer to my problem is here: https://www.anaconda.com/blog/utilizing-the-new-compilers-in-anaconda-distribution-5. The install process isn't compatible with the most up-to-date XCode, and the user needs to get the 10.9 SDK off Github as described there, then point the compiler at it with "export CONDA_BUILD_SYSROOT=/Opt/MacOSX10.9.sdk" (assuming it's downloaded to Opt).]

(I should note the above broadly duplicates an issue I have raised directly on Github.)

The current install instructions appear to assume GDAL is installed on the target machine, and do not say so. They must make this explicit, as the triggered install errors are extremely opaque. I gather from asking on Github that these errors do not actually prevent install, but the readme needs to provide that guidance because the errors look pretty scary – and also should I think recommend a GDAL install anyway. [I got GDAL in a pain-free fashion using homebrew on my Mac (https://brew.sh), if that helps – or just let users figure it out for themselves.]

There should also be a line in the readme telling the user how to launch the software

after install, as there is over at r-barnes/Barnes2019-DepressionHierarchy.

My familiarity was C++ testing is not very complete, but I don't believe this code has a *formal* testing framework attached to it. This is more an observation than anything else, but I would informally recommend that the authors consider building one at some point in the future. This is the way the wind seems to be blowing in research software development at the moment, and it might be good to get ahead of the pack on this.

On the basis that the code is not formally tested, I have not attempted to extensively review the code itself. However, informal tests are applied in the manuscript and in the code (the correctness tests), and this for me meets the basic requirements for a submission like this. Could you add a line in the readme telling the user how to launch the correctness tests for themselves...? This would also serve as a nice example of the code doing its thing.

As claimed in the manuscript, the code is indeed well commented and neatly and nicely structured – good job!

Code availability statement is great – but you should consider actually putting an open source licence onto the code and making that explicit here (after all, this is already de-facto open source code, so you should probably formalise it). E.g. add a LICENSE.txt and add "The code is released under an MIT Open Source license [or your preferred license]" at the end of this section. But your call.

---

## Author Comment (AC1) · 18 Sep 2020

**Interactive comment on Computing water flow through complex landscapes, Part 3:**
**Fill-Spill-Merge: Flow routing in depression hierarchies**

Richard Barnes[1,2,3], Kerry L. Callaghan[4,5], and Andrew D. Wickert[4,5]

[1]Energy & Resources Group (ERG), University of California, Berkeley, USA
[2]Electrical Engineering & Computer Science, University of California, Berkeley, USA
[3]Berkeley Institute for Data Science (BIDS), University of California, Berkeley, USA
[4]Department of Earth Sciences, University of Minnesota, Minneapolis, MN, USA
[5]Saint Anthony Falls Laboratory, University of Minnesota, Minneapolis, MN, USA

**Correspondence:** Richard Barnes (richard.barnes@berkeley.edu)

**1 Anonymous ref 1:**

**Comment:** figure 1: - I suggest to number the depressions in the figure to make it easier for the reader to follow the explanations.

**Response:** Thank you for the suggestion, we've added numberings to the depressions.

**Comment:** lines 70–83: I think that if the authors refer to the subfigures it will be easier to read. I found that fig. 2e has the elements I needed to understand the text.

**Response:** Thank you for this suggestion, we've added subfigure references to Figure 2 throughout the paper, except in cases where it seems as though referencing the entirety of the figure is more appropriate.

**Comment:** line 78: when #0 is mentioned in fig. 2, I had a bit of trouble finding it in the figure (it's a bit hidden under "Ocean", and the purple color of the square in 2a is a bit dark).

**Response:** We've left 0 under the ocean, but have changed the text in the purple square to white.

**Comment:** line 93: figure 2e; line 100: figure 2a

**Response:** We have added these specific figure numbers. The sentence in question previously read

[...] margin of water contained within the parent (i.e., the "marginal volume" indicated on Figure 2).

It now reads

[...] margin of water contained within the parent (i.e., the "marginal volume" indicated on Figure 2e).

and

These are the dashed lines shown in Figure 2.

now reads

These are the dashed lines shown in Figure 2a

**Comment:** line 173: I was a bit confused by the "water than they can hold spill" here. Is it "water than they can hold will spill"?

**Response:** The sentence previously read

Water must be redistributed such that leaf depressions containing more water than they can hold spill over into their neighboring depression.

We have rephrased this sentence to clarify:

When a leaf depression initially contains more water than it can hold, the water will be redistributed by spilling over into the neighboring depression.

**Comment:** line 382: maybe insert a citation for GRASS GIS here? (Neteler et al 2012)

**Response:** We have added the requested reference. The text previously read

[...] for analysis and manually removed several road bridges using GRASS GIS to prevent artificial pooling behind these [...]

It now reads

for analysis and manually removed several road bridges using GRASS GIS (Neteler et al., 2012) to prevent artificial pooling behind these [...]

**Comment:** figures 6 and 7: Have you thought about using a divergent colorscale here? Since you are showing positive and negative differences?

**Response:** We considered several colorscale options, including a divergent colorscale, in light of this comment. After comparing several options, we decided that a continuous colorscale with discrete classes was most effective, since this allows a viewer to more easily differentiate positive and negative values. Since the values in question are generally small, we also switched to a logarithmic scaling of the colorbar.

**Comment:** Is the drainage network from both algorithms compared here are identical, differing only at the depressions? I was curious to see the drainage.

**Response:** We have included figures 1–4 showing the drainage networks from both algorithms in this response, but have not added these to the paper because we did not think that these added significant value to the paper. The drainage networks are extremely similar between the two sets of results, with minor differences at locations where the depression fills differ.

[Figure]

**Figure 1.** Comparison between drainage networks created over a surface filled using FlowFill versus a surface filled using FSM in the Sangamon River basin. Results appear close to identical for 0.1 m of runoff. With 0.05 m of runoff, minor differences are visible in the upper left portion of the figure.

[Figure]

**Figure 2.** Comparison between drainage networks created over a surface filled using FlowFill versus a surface filled using FSM in the Sangamon River basin. Results appear near identical for 0.01 m and 0.001 m of runoff.

**2 Ref 2, Daniel Hobley**

**Comment:** I found the way the depression hierarchy is talked about in the Algorithm section quite confusing. I think the reason is that the authors convolve the object that is the hierarchy itself (i.e., the input to this algorithm), the conceptual idea of the DH, and the method used to build that hierarchy (as explained in Barnes et al. (2020)). For example

- around P5ln30 "The specific outputs from the depression hierarchy [the method] that are used in the Fill-Spill-Merge algorithm are: — DH: the depression hierarchy [the object] itself....";

- P8ln2 "By routing water according to the DH..."

So is this the object, or the method? Finding a distinct way to refer to the method/algorithm that produces a DH, the ordering that is described by a DH, and a DH structure itself would significantly enhance clarity throughout. (See line item P9ln26 also.)

[Figure]

**Figure 3.** Comparison between drainage networks created over a surface filled using FlowFill versus a surface filled using FSM in the Río Toro basin. Results appear near identical for 15 m and 5 m of runoff.

**Response:** Thank you for pointing this out. We have normalized the terminology so that "depression hierarchy" should always refer to the data structure itself. We feel that the data structure and the conceptual idea of the DH are closely aligned and so do not distinguish explicitly between them. However, we now make it clear when we are refer to the algorithms or construction of the DH.

[Figure]

**Figure 4.** Comparison between drainage networks created over a surface filled using FlowFill versus a surface filled using FSM in the Río Toro basin. Results appear near identical for 1 m and 0.2 m of runoff.

**Comment:** Jargon suddenly appears at the end of p. 9. The whole section P9ln25–P10ln17 I found very confusing. I ramble about this at length in the line comments below, but at heart this needs a. the jargon explaining, b. the text simplifying, and c. the text expanding as necessary.

**Response:** As detailed below, we have added additional background material and citations to the paper to help explain the terminology. We have also gone through and tried to clarify the wordings.

> **Comment:** Regardless of that, I would strongly advocate that you put some actual pseudocode into the paper – maybe as supplemental material, but in the body would also be good. This would really help in sections like that one, and the one to follow (both noted below).

**Response:** We had meant the source code accompanying the paper to serve as an additional reference and now state this explicitly in the paper:

> This paper is also accompanied by complete, well-commented source code; the reader may find it helpful to download this code and refer to it as an additional reference.

We have also included pseudocode for the nested traversals, since they seemed particularly problematic. This is included at the end of the paper and referred to at various points in the text.

> **Comment:** Similarly, if iterations like those on P10 are described, then I think they would be much clearer with cartoons illustrating the possible steps next to them. You use fig 2 as a case study for the sequence on P11, but it would be much clearer to produce a second version along the lines of fig 2 but with multiple parts. These would highlight each inner loop as described in the text, what happens under the possible configurations of that loop, and the sequencing of the water filling in under the rules for those configurations.

**Response:** We considered creating a figure to highlight the inner and outer loops of the traversal, but decided against it for two reasons:

1. The pseudocode, written in response to your above comment, as well as our revised text, significantly clarify the outer and inner traversals and their roles.

2. The traversals are important to the algorithmic approach, but are not essential for a conceptual understanding of how FSM moves water across a landscape.

As a result of these considerations, we thought that the best layering of information would be a figure to help guide the reader through the essential conceptualization of the flow-routing process (cf. especially Fig. 2 in the revised manuscript), with the improved textual description and pseudocode available to help the reader curious to understand the inner workings of the algorithm. We hope that this nonetheless addresses the spirit of this comment.

40

**Comment:** Oceanlink-geolink terminology. I like the conceptual framework of the paper a lot, but the description of "paired oceanlink-geolinks" I found confusing. Because there can never be an oceanlink without a geolink, I would tweak the terminology such that an oceanlink is a subset of a geolink, and the pair concept goes away.

**Response:** We now separate the concepts of geolinks and oceanlinks entirely. Geolinks move water within a binary tree and oceanlinks move water between binary trees.

**Comment:** The maths as described on P13 is not strictly correct. The equations give summed elevations, but they are variously described as "capacity" or "volume". There should be cell areas in here somewhere. Also, there is a missing definition of $N$. Check the maths again for precision of terms.

**Response:** Thank you for this comment. We now define $N$, as requested. The other difficulty arose because we implicitly assumed that the problem was scaled such that cells have an area of 1. We have now generalized the math to cells of arbitrary area before showing this special case.

**Comment:** Section 5 – The text asserts that $O(N \log N)$ is the best fit, but the paper does not show it. At the very least, plot the data in Table 1...however, doing this will reveal that the data points do not in fact give a good smooth correlation, convincing of $N \log N$ (which is presumably why you didn't do it in the first place!) I would retain Table 1 as nice examples of "real world" model performance, but add a synthetic test with random noise elevation on varying size grids. This should give you the convincing $N \log N$ scaling to plot up and push a $N \log N$ scaling best fit through, and is also a cleaner — and more standard - way making the point about actual performance of your implementation. Some light restructuring (see below) would help here too.

45

**Response:** We have added an addition figure showing how the performance of the algorithm scales on synthetically generated

datasets. We also add an additional paragraph:

> To more precisely demonstrate performance, we run Fill-Spill-Merge on synthetic landscapes of various sizes generated using RichDEM's Perlin noise random terrain generator. Multiple landscapes are generated and timed at each size to smooth timing variation due to both the data and fluctuations in the testing environment. This results in Figure X, which again shows that the performance data gives a good fit to an $N \log N$ function.

**Comment:** You could add a statement somewhere that this algorithm also permits water to be retained in the depressions between runs, if that is desired – so you can get a timeseries as the lakes fill over time. It would also easily let you perform true water balances on the lakes if the user wanted to, by subtracting water between steps. This is implicit in the text, but it's a cool feature you could emphasise a bit more if you want.

**Response:** Thank you for this suggestion, we have added several notes about this.

– §3: "Note that the landscape may already have standing water at this stage."

– §3.1: "(perhaps referring to a matrix of rainfall values, but also existing stores of standing water)"

– §6: "If standing water is retained between invocations of Fill-Spill-Merge, and new water added at each invocation, the algorithm can be used to simulate the movement of water across landscapes; we will explore this further in future work."

**Comment:** P1ln16 – misformatted ref

50

**Response:** Fixed, thank you. The sentence previously read

> [...] and cratering Cabrol and Grin (1999).

It now reads

> [...] and cratering (Cabrol and Grin, 1999).

**Comment:** P3ln15 — 0 seems a special case here. Refer also to 12 or 11?

**Response:** We believe you're referring to the definition of root. The sentence previously read

Root: A depression, such as #0 in Figure 2, that has no parent. This term may also refer to any node that is used as the starting point for a traversal that only considers the node and its descendants.

It now reads

Root: A depression, such as #0, #11, and #12 in Figure 2, that has no parent. This term may also refer to any node that is used as the starting point for a traversal that only considers the node and its descendants.

**Comment:** P5ln10 – The ocean links. You should make it clear here why, from a modelling perspective, you need to separate ocean links from geolinks, so we know where we're going. Also, you have inconsistency between "ocean link" and "oceanlink" – check for consistency (incl. In figure captions).

**Response:** To address this comment we have added a new paragraph to the end of §2:

Within the algorithm, oceanlinks and geolinks are used for different purposes: an oceanlink tells us that the depression in question has grafted onto the leaf node of another tree of the depression hierarchy, locating a route for overflowing water to eventually reach the ocean. The depression to which it is oceanlinked is considered its parent, but it is not the child of that depression because water flows only one way along an oceanlink. In Figure 2a, depression #4 can be considered the parent of #12, but #12 is not the child of #4. This is because #12 is not physically contained within #4, but #12 will send all of its overflowing water to #4, as shown in Figure 2b–e. #4 will not contain the total water volume contained within #12, unlike other parents. Geolinks are more general: not every depression has an oceanlink, but every depression has a geolink. The oceanlink provides information about the type of overflow happening, but it is the geolink that will be used to track the actual direction of spill of water in the depression hierarchy.

55   We have normalized to the term "oceanlink".

**Comment:** Fig 4. This caption could be a lot more generous to the reader. Explain the colorcoding; what *exactly* the blue shading is; and exactly what you mean by "non-additive". I don't quite follow, and feel there must be a clearer way of expressing this. Also, would it not be clearer to actually let area represent volume of water? (I assume that this is not the case is what you are getting at by "non-additive"?) In particular g–j are quite hard to follow and need more text. Isn't j inconsistent with f? You appear to suggest in f the whole system floods out, but j does not show this. Format-wise, you have a,b,c on the fig and A,B,C in the text.

**Response:** We adjusted this figure to make the height of blue bars represent an actual volume of water. This removes the

need for the 'non-additive' comment, and should reduce confusion in interpretation of the figure. We have also changed the caption to add more information. The caption used to read:

**Visual Overview of the Algorithm.** In this figure the heights of the water bars are non-additive: only the changes between panels are important. The algorithm consists of three major stages (Figure 5). From its initial distribution (A), water is moved downhill into pit cells (B, §3.1). Water is then moved within the depression hierarchy (C–F, §3.2): water in depressions with insufficient volume overflows first into their sibling depressions (D) and then – if the sibling depression becomes filled – passes to their parents (E, F). Any leftover water overflows into the ocean (F) and is forgotten. Depressions to be flooded are then identified and flooded (§3.3) starting from an arbitrarily-chosen pit cell (G–J).

It now reads:

**Visual Overview of the Algorithm.** Black outlines represent the elevations of the cells. Blue areas are the heights of water in each cell or depression within the depression hierarchy. Capital letters label cells, and numbers on colored dots label depressions. Colors at the base of each panel match the colored dots and indicate to which depression each cell belongs. The algorithm consists of three major stages (Figure 5). From its initial distribution (a), water is moved downhill following flow directions in the steepest downslope direction from each cell, as indicated by the arrows. Water continues to move downslope until it reaches the pit cells (b, §3.1). Water is then moved within the depression hierarchy (c–f, §3.2). (c) shows the initial distribution of water within the depression hierarchy, based on how much water was in the pit cell of each depression. Water in depressions with insufficient volume overflow first into their sibling depressions and then – if the sibling depression becomes filled – passes to their parents. All of the leaf depressions in (c) are completely filled, so no sibling depressions can accommodate more water. Therefore, depressions 1 and 2 pass their overflowing water up to their parent, depression 6, and depressions 3 and 4 pass their overflowing water up to their parent, depression 5 (d). Depression 6 is now overflowing, but its sibling, depression 5, is not full, so depression 6 passes as much of its overflowing water as it can to depression 5 (e). Once depression 5 is full, some overflowing water still remains, so this is passed to the parent, depression 7 (f). In this case, depression 7 is able to accommodate the remainder of the water. Had depression 7 also overflowed, the leftover water would have overflowed into the ocean and been disregarded. Depressions to be flooded are then identified and flooded (§3.3). Since depression 7 contains water, we know that all of its descendants must be completely full. Therefore, we can flood these all at the same time, on the level of depression 7. Any one of the pit cells within depression 7 is arbitrarily selected as the starting point (g). More cells are added until all of the water has been accommodated. (h-j) are a visual representation of this process, although the algorithm would first locate affected cells C-J, and then calculate the final height of water in all of these cells in a single step.

**Comment:** P9ln10 – "depth/breadth-first traversal". Please explain what this is in plainer English.

**Response:** These terms are fundamental to the study of algorithms and would be taught in an intro level course, as such we think it's alright to assume reader familiarity. We now state this directly at the beginning of §2.

> Many of the techniques in this paper are based on binary tree data structures and their traversals. Although we define terms below, more complete explanations and visual examples can be found in the text for any introductory undergraduate course on data structures. We recommend Skiena (2008) and Sedgewick and Wayne (2011) as good references. In particular, a good understanding of recursion will be helpful.

Further, we add to the end of §2:

> With these linkages in place, we can consider various ways of traversing the trees. Given a binary tree $T$ with left and right children $T.L$ and $T.R$, a breadth-first traversal considers both $T.L$ and $T.R$ before considering any of $T.L.L$, $T.L.R$, $T.R.L$, or $T.R.R$. A depth-first traversal, on the other hand, will consider $T.L$ and all of its descendants before considering $T.R$ or any of its descendants. The tree traversals we perform in this paper are all depth-first.

We now note in the text that an invariant is "(a property which is true before and after each call a function)".

This page demonstrates a use of invariants to prove the correctness of insertion sort: http://www.cs.xu.edu/csci220/01f/ insertionProof.html.

**Comment:** P9ln25 – Suddenly, the paper becomes much more jargon-y and hard to follow. Clarifying depth-first at ln 10 will help, but what is "post-order"? "an invariant"? Please explain these in plainer English. This paragraph and the next are also quite confusing aside from the jargon, so please expand and be more generous with the text as well.

**Response:** Again, this terminology is standard to introductory algorithms. However, to the end of §2 we add:

Depth-first traversals are most naturally expressed via recursion and come in three types: in-order, pre-order, and post-order. Let a recursive traversal function be called $r(\cdot)$ and the processing we perform on a particular node in the tree $p(.)$, then the traversals are given by:

- in-order: $r(T.L)$ then $p(T)$ then $r(T.R)$

- pre-order: $p(T)$ then $r(T.L)$ then $r(T.R)$

- post-order: $r(T.L)$ then $r(T.R)$ then $p(T)$

**Comment:** Things I am struggling with: How does a depth-first traversal have an "outer" rather than a "deeper"? Clearly the water starts in the leaves, so how can the outer (first??) traversal begin in the ocean (ln 30), which is a trunk? How are the traversals nested, and when do we shift from outer to inner and vice versa? An explanation of the jargon in a pre-paragraph will probably sort most of this – i.e., what are outer and inner traversals, what is pre- and post-order, and how are these concepts interrelated? Another flowchart, a diagram, or chunk of pseudocode would also really help here. The steps on P10 help piece together what these paragraphs must mean, but they come after the confusing text not with it, and the text should really stand on its own.

65

**Response:** To address this we've added a new section ("Traversals") more clearly defining the different traversals that can

be used on binary trees.

We have also added and modified the language in this section:

> To effect the intuition developed above, we need a well-defined way to visit all of the nodes in the depression hierarchy. A post-order traversal allows us to visit both of a node's children and their descendants before calculating any quantities on the node itself. The result is that leaves get processed before their parents. However, a single traversal is insufficient: we need one traversal (the "outer" traversal) to identify nodes that have excess water and another traversal (the "inner traversal") to distribute this water. The outer traversal may launch the inner traversal many times as it works its way up hierarchy.

> To efficiently redistribute water, the Fill-Spill-Merge algorithm performs nested depth-first traversals of the DH. The outer traversal is post-order and considers each meta-depression in turn, from the most deeply nested to the least. For each meta-depression, an inner traversal handles its overflows by moving water to its sibling (starting by filling the sibling's descendants) and, if there's any left, passing it to the depression's parent. In this way, the outer traversal maintains an invariant (a property which is true before and after each call a function): any meta-depression it has processed does not contain an overflow. Put another way, the outer traversal finds problems and the inner traversal fixes them.

We have added pseudocode for the traversals and refer to it at appropriate points in the text.

**Comment:** P10ln10 – "We add water *from A* to B"? Where does the added water come from? Is it the original volume or the passed volume? (Must be the latter, but this is not as written) Tweak the phrasing here. P10ln11 – "this part of the algorithm". Do you mean point 1, the inner traversal, or the section 3.2? Point (iii) in this list is also very confusing. Break this case out and say what you mean (and why this is a terminating case) more clearly. The manuscript has not described the concept of an oceanlink-geolink pair (surely the link between two nodes is either an oceanlink or a geolink, so there can be no pairing?), so clarify that too (see main comment). Again, this feels like a diagram or pseudocode or both would add a lot here. The diagram I have in mind would show the possible cases for this 1-3 loop as a sequence of filling lakes, as in Fig 2 or fig 4a,b, g-j (but where water volume is honoured). Although I appreciate this section looks like this because it explicitly mirrors the structure of the code, there is surely a clearer way to express 1-3. E.g.– 1. If a node has excess water, attempt to pass that water to an unfilled sibling.– 2. If no unfilled sibling is available, pass the water to the parent and terminate inner loop.– 3. If no parent exists, the node must have an oceanlink, so pass the water along the oceanlink and terminate inner loop.– 4. If the oceanlink ends in the ocean, discard the excess water and terminate inner loop. Does this not encompass all the logic without the detail in the subcases, the pseudo-parents, etc.?

70  **Response:** We have tried to clarify the wording here as follows

When an overfilled depression is located by the outer traversal above, its water needs to be redistributed to neighbouring depressions. If we call the overfilled depression $D$, then the water can be redistributed by starting a second, inner post-order traversal at $D$. This inner traversal carries Excess Water from one depression to another until it has found a home for all of it. When we pass water into a depression, it can go to one of three places: the depression itself, its sibling, or its parent. Distributing the water to any of these places may itself cause an overflow. Therefore, the inner (pre-order) traversal comprises the following steps:

1. Call the depression that we are currently considering $B$. This may be the depression we originally considered, depression $D$, or it may be some other depression reached during the steps detailed below. If $B$ is overflowing, we add the overflow to the Excess Water the inner traversal is carrying. If $B$ has spare capacity we add water from the Excess to $B$ until either it fills or all of the Excess Water the inner traversal is carrying is used.

2. At this point, the inner traversal can terminate if: (i) there is no water left, (ii) $B$ is the parent of $D$, or (iii) $B$ acts as a parent of A by receiving its overflow via an oceanlink.

. . .

We have added pseudocode for the traversals and refer to it at appropriate points in the text.

**Comment:** P10ln31 – apart from still disliking the "paired" terminology here, this also brings back the pt (iii) I didn't understand above. This makes it clear what you meant there, but the idea that 12 is sort of a parent of 4, but not really, and only sometimes, is a confusing one. I would stick with the idea that parents are strictly in the same tree, and replace this idea with simply "passing water down the oceanlink to move it between trees". I don't see what else it is adding for the cost of being confusing. In general though, this stuff on P11 is really good.

**Response:** As above, we've tried to better separate the concepts of geolink and oceanlink. We've also tried to simplify the

language in this section to separate the concepts of parent and oceanlink:

1. Call the depression that we are currently considering $B$. This may be the depression we originally considered, depression $D$, or it may be some other depression reached during the steps detailed below. If $B$ is overflowing, we add the overflow to the Excess Water the inner traversal is carrying. If $B$ has spare capacity we add water from the Excess to $B$ until either it fills or all of the Excess Water the inner traversal is carrying is used.

2. At this point, the inner traversal can terminate if: (i) there is no water left, (ii) $B$ is the parent of $D$, or (iii) $B$ was reached via an oceanlink.

3. Otherwise, if $B$ has a sibling and the sibling's water volume is less than its depression volume, then start from Step 1 with the new $B$ set as the depression pointed to by the current $B$'s geolink.

4. Otherwise, if $B$ has no sibling or the sibling's water volume is equal to its depression volume, then start from Step 1 with the new $B$ set as the parent of the current $B$ or, if $B$ has no parent, then whatever depression $B$ oceanlinks to.
* * *
**Comment:** P12lns3-4 – I don't see why the algorithm needs to start at the ocean and iterate back to every leaf, only to iterate forward again. Given we already maintain a list of all the leaves, associated pits, and labels, why can't me just start at the leaves and only go forward? If I'm missing something, it means this needs clarifying anyway. Are you trying to say, iterate from the ocean down along all the branches, but pinch out the branch whenever you find water and don't go further?

75
* * *
**Response:** As noted in §2, "a depression hierarchy (DH) is a data structure representing a forest of binary trees."

There are no lists of leaves or pits: this information is all organized into trees. The only way to find the leaves is by traversing the tree. The only way to move information around in the tree is via (recursive) depth-first traversals. Iteration, in the strict sense of the word, doesn't apply in this context.

It is possible we could design a different algorithm that somehow tracks leaves and moves from them towards the ocean via a breadth-first traversal, but this would likely be more difficult to describe, less intuitive, and certainly more difficult to implement.

**Comment:** P12lns15-28: Another place where a block of pseudocode alongside the text would add a lot. Arguably it could replace the text entirely, but having both would be extra clear.

**Response:** We have added pseudocode for this function as well and references to it in the text.

**Comment:** P12ln29. "To do so, we imagine a hypothetical outlet that drains the depression..." add: "at this elevation". This section for me says the same thing three times in different words. I would have simply: "Step 1 in this approach requires an efficient way to determine the volume of a depression below any given elevation. To do so, we imagine a hypothetical outlet that drains the depression at that elevation. If we call the elevation of this hypothetical outlet o and..." (also see comment below about maybe removing the idea of a hypothetical outlet entirely.)

**Response:** Thank you, we appreciate your wording suggestion here and below and have adopted the strategy you suggest. This section used to read:

Step 1 in this approach requires an efficient way to determine the volume of a depression below any given elevation. To do so, we imagine a hypothetical outlet that drains the depression. If the depression is full enough that of its all cells receive water, then the elevation of this hypothetical outlet is simply that of the topographic outlet from the depression. If the depression is not yet completely filled, it can be visualized as a pipe in the side of the depression that is an infinite sink for any water entering it, thereby acting analogously to an overflow drain below the edge of a sink or bathtub. If we call the elevation of this hypothetical outlet $o$ and a depression contains $N$ cells of elevations $\{a, b, c, d, \ldots\}$, then the total volume per unit area (i.e., height) of water that the depression can accommodate is

It now reads:

Step 1 in this approach requires an efficient way to determine the volume of a depression below any given elevation. If we call this elevation $o$ and the depression below the outlet contains $N$ cells with elevations $\{a, b, c, d, \ldots\}$, then the total volume per unit area (i.e., height) of water that the depression can accommodate is

80

**Comment:** Equation 1: You don't actually define N

**Response:** We have now defined N:

If we call the elevation of this hypothetical outlet $o$ and a depression contains $N$ cells of elevations $\{a, b, c, d, \ldots\}$

**Comment:** Equation 3: I don't object to it, but why is there a bar through V?

**Response:** We have barred the bar from the paper.

**Comment:** P13ln4 – Run-on; end with a period not comma or tweak phrasing.

**Response:** This sentence previously read

Now, consider cells $c_i = c_1, \ldots, c_N$ in the plain queue (i.e., those that have been visited and popped from the priority queue), we can calculate the volume of the depression below that of the last cell popped from the priority queue, the sill $z_s$, as: [EQU] Here, $V_{dep,z_s}$ is the volume of the depression below $z_s$, and $z_i$ is the elevation of cell $c_i$.

It now reads

Now, consider cells $c_i = c_1, \ldots, c_N$ in the plain queue; that is, those cells that have been visited and popped from the priority queue. We can calculate the volume of water that can be accommodated in the depression below the elevation $z_s$ of the last cell $c_N$ (the sill) as: [EQU] where $z_i$ is the elevation of cell $c_i$ and $a_i$ is the area of cell $c_i$.

**Comment:** P13ln5 – "the sill". Which? The hypothetical one, or the real one? In general here, I think the concept of an "imaginary" outlet at every possible node height is getting in the way of what you're trying to say. Isn't it just that you can track the volume of a depression below various depths using the equations you give by cumulative summing along the plain queue, and then can compare that increasing volume as you move along the queue to the water volume available, which tells you the fill level? I would rephrase the way you present P12ln29–P13ln9 without this imaginary sill/outlet concept.

**Response:** We've eliminated the "imaginary" outlet in accordance with your suggestion, thank you. (See also, above.)

**Comment:** P14ln32 — If you rename the section, I don't think this final sentence is needed.

90 **Response:** We feel the sentence contributes to the flow and so have opted to retain it, but we appreciate the suggestion.

**Comment:** P15ln11 – consider "Applications and testing on real-world data" or words to that effect (assuming you renamed 4 above, and remove the subheading below).

**Response:** We have renamed the section per your suggestion.

**Comment:** P13ln21 – I don't think this is the right heading for the section. Isn't this "Algorithm performance" or something like that more specific? 4 and 5 would then be subsections,e.g., "theory", "computational performance". The section itself reads really well though — and I assume by this point you will have defined the iteration jargon a bit better at first use.

P16ln1 – there is no 6.2!? Remove the subheading. Otherwise, I like this section a lot....overall, I think a light restructuring (i.e., broadly just some section renaming and text shunting) through 4-6 would make this a bit stronger. What you have is testing of performance on both synthetic (please!) and real-world data, then a real-world case-study that attempts some model validation. I think the sectioning should reflect that a bit more explicitly, e.g. something like,4. Model performance– 4.1 Theoretical analysis– 4.2 Performance on synthetic data 5. Field testing and inter-model comparison– 5.1 Performance on real world data(brief chat about Table 1 to give people an idea of general performance on real-world data, or just remove and junk these subsections)– 5.2 Model intercomparison (This is basically just 6.1 – and by all means, keep the performance data for this case study within this section)

**Response:** Thank you for this suggestion, we have organized these sections under "Algorithm performance" with subsections for "Theory", "Computational Performance", and "Model Intercomparison". We have also added performance tests on synthetic data.

**Comment:** (Technical) Install instructions in the readme require cmake – and so should probably explicitly direct the user to acquire this first. I note configuration issues are also likely, which should probably also be clarified (see below); if they are not, the authors are dramatically reducing the potential reach and impact of their work. In general, I think the readme needs expanding a bit (see technical comments).

**Response:** We make a note that cmake is required in `7929c96a3832e819c46e3812618a118681467852`. Configuration issues are always a possibility, but the use of cmake and the opportunity for users to raise configuration issues via Github should reduce this potential.

**Comment:** (Technical) I severely struggled with the install for the software, and am probably significantly more tech-savvy than many target users. Did the authors test installs thoroughly against a full range of OSes? I am running macOS Catalina, XCode 11.4 (i.e., fully modern installs) and anaconda, and am seeing a spew of cmake errors which some investigation reveals are related to incompatibilities between the installer, the shipped SDKswith XCode, and anaconda. This is going to be a common user case, and I think the authors need to address it, if only briefly, in the docs – and test some other common environments too for safety's sake. [Part of the answer to my problem is here: https://www.anaconda.com/blog/utilizing-the-new-compilers-in-anaconda-distribution-5. The install process isn't compatible with the most up-to-date XCode, and the user needs to get the 10.9SDK off Github as described there, then point the compiler at it with "export CONDA_BUILD_SYSROOT=/Opt/MacOSX10.9.sdk" (assuming it's downloaded to Opt).] (I should note the above broadly duplicates an issue I have raised directly on Github.)

**Response:** We've tested the code on the range of systems we have access to and are responsive to issues raised on Github. The reviewer's particular issues have been addressed at:

– https://github.com/r-barnes/Barnes2020-FillSpillMerge/issues/3

– https://github.com/r-barnes/Barnes2020-FillSpillMerge/issues/5

We have also removed the NetCDF dependency.

**Comment:** (Technical) The current install instructions appear to assume GDAL is installed on the target machine, and do not say so. They must make this explicit, as the triggered install errors are extremely opaque. I gather from asking on Github that these errors do not actually prevent install, but the readme needs to provide that guidance because the errors look pretty scary – and also should I think recommend a GDAL install anyway. [I got GDAL in a pain-free fashion using homebrew on my Mac (https://brew.sh), if that helps – or just let users figure it out for themselves.]

**Response:** GDAL is not required by either algorithm, but is used by the demonstration/example code. We now clarify this in the cmake files and README. Per the Issues you raised on Github (see above), we now provide Mac installation instructions in the README.

100

**Comment:** (Technical) There should also be a line in the readme telling the user how to launch the software after install, as there is over at r-barnes/Barnes2019-DepressionHierarchy.

**Response:** The README now contains the following:

A test program, `fsm.exe` is generated by make. This program simulates pouring a given amount of water onto every cell on a landscape and determining where it all ends up. Running the program shows its command-line arguments.

**Comment:** (Technical) My familiarity was C++ testing is not very complete, but I don't believe this code has a \*formal\* testing framework attached to it. This is more an observation than anything else, but I would informally recommend that the authors consider building one at some point in the future. This is the way the wind seems to be blowing in research software development at the moment, and it might be good to get ahead of the pack on this. On the basis that the code is not formally tested, I have not attempted to extensively review the code itself. However, informal tests are applied in the manuscript and in the code (the correctness tests), and this for me meets the basic requirements for a submission like this. Could you add a line in the readme telling the user how to launch the correctness tests for themselves...? This would also serve as a nice example of the code doing its thing.

**Response:** For this revision, we have substantially expanded the tests accompanying our code. The README file explains how to compile and run these tests. We use *random testing* in conjunction with known reductions (such as depression filling) and multiple-paths analysis (such as progressively adding small amounts of water to a landscape versus adding a large amount all at once) to perform both unit tests as well as end-to-end tests on the code. As of `a350c410cafbb4caef4`, our test suite subjects our code to 208,490 distributed over 11 test cases; all assertions pass.

**Comment:** (Technical) Code availability statement is great – but you should consider actually putting an open source licence onto the code and making that explicit here (after all, this is already defacto open source code, so you should probably formalise it). E.g. add a LICENSE.txt and add "The code is released under an MIT Open Source license [or your preferred license]" at the end of this section. But your call.

105

**Response:** Thank you for the suggestion, we have moved to the MIT license in `a28264bd337975e5fa4e335c03`.

**3   Other:**

While adding additional tests to the code in response to the reviewers' requests we discovered a bug in Fill-Spill-Merge that had caused some discrepancies in the results in our original submission. We have since corrected this bug, and reran the model

110 to obtain correct results. The figures in the "Model Intercomparison" section have been changed to show these updated results; they also incorporate a new colour scheme based on the reviewers' requests.